# Brazilian germplasm of winter squash (*Cucurbita moschata* D.) displays vast genetic variability, allowing identification of promising genotypes for agro-morphological traits

**Ronaldo Silva Gomes**[1]*, **Ronaldo Machado Júnior**[1], **Cleverson Freitas de Almeida**[1], **Rafael Ravaneli Chagas**[1], **Rebeca Lourenço de Oliveira**[1], **Fabio Teixeira Delazari**[2], **Derly José Henriques da Silva**[1]

**1** Agronomy Department, Federal University of Viçosa-UFV, Viçosa, MG, Brazil, **2** National Service of Rural Learning- SENAR, Campo Grande, MS, Brazil

* ronaldo.s.gomes@ufv.br

## Abstract

Winter squash fruits (*Cucurbita moschata* D.) are among the best sources of vitamin A precursors and constitute sources of bioactive components such as phenolic compounds and flavonoids. Approximately 70% of *C. moschata* seed oil is made up of unsaturated fatty acids, with high levels of monounsaturated fatty acids and components such as vitamin E and carotenoids, which represent a promising nutritional aspect in the production of this vegetable. *C. moschata* germplasm expresses high genetic variability, especially in Brazil. We assessed 91 *C. moschata* accessions, from different regions of Brazil, and maintained at the Federal University of Viçosa (UFV) Vegetable Germplasm Bank, to identify early-flowering accessions with high levels of carotenoids in the fruit pulp and high yields of seed and seed oil. Results showed that the accessions have high variability in the number and mass of seeds per fruit, number of accumulated degree-days for flowering, total carotenoid content, and fruit productivity, which allowed selection for considerable gains in these characteristics. Analysis of the correlation between these characteristics provided information that will assist in selection to improve this crop. Cluster analysis resulted in the formation of 16 groups, confirming the variability of the accessions. *Per se* analysis identified accessions BGH-6749, BGH-5639, and BGH-219 as those with the earliest flowering. Accessions BGH-5455A and BGH-5598A had the highest carotenoid content, with averages greater than 170.00 μg g$^{-1}$ of fresh mass. With a productivity of 0.13 t ha$^{-1}$, accessions BGH-5485A, BGH-4610A, and BGH-5472A were the most promising for seed oil production. These last two accessions corresponded to those with higher seed productivity, averaging 0.58 and 0.54 t ha$^{-1}$, respectively. This study confirms the high potential of this germplasm for use in breeding for promotion of earlier flowering and increase in total content of fruit pulp carotenoids and in seed and seed oil productivity.

**Data Availability Statement:** All relevant data are within the paper and its Supporting Information files.

**Funding:** All the funding sources of support to this study corresponded to study scholarships received by the first author (Ronaldo Silva Gomes). The first scholarship corresponded to a master's scholarship (grant number 001), funded by the Coordination for the Improvement of Higher Education Personnel (CAPES). The second scholarship corresponded to a doctorate scholarship (doctorate-GD grant), funded by the National Council of Technological and Scientific Development (CNPq). There was no additional external funding received for this study.

**Competing interests:** The authors have declared that no competing interests exist.

# Introduction

Winter squash (*Cucurbita moschata* D.) is one of the vegetables of greater socio-economic importance in the *Cucurbita* genus, largely due to the high nutritional value of its fruits and seeds. The pulp of its fruits constitutes an important source of carotenoids such as $\beta$-carotene, the precursor of greater pro-vitamin A activity [1, 2, 3]. The pulp is also an excellent source of minerals such as K, Ca, P, Mg, and Cu [4, 5]. The socio-economic importance of *C. moschata* is also linked to the high volume and value of its production. It is estimated that, together with other cucurbits such as *C. pepo* and *C. maxima*, the cultivated area and the world production of this vegetable in 2017 were approximately 2 million hectares and 25 million tons, respectively [6], most of it concentrated in China and India. In Brazil, this crop is of high socio-economic importance, with a cultivated area of approximately 90 thousand hectares, an estimated production of more than 40 thousand tons / year, and an annual production value of around R$ 1.5 million [7].

*C. moschata* brings together characteristics that are fundamental to biofortification programmes, such as high productivity and profitability potentials, high efficiency in reducing micronutrient deficiencies in humans, and good acceptability by producers and consumers in regions where it is grown [8]. This has caused this vegetable to be chosen as a strategic crop for breeding programmes promoting biofortification, such as the Brazilian Biofortification Program (BioFORT), led by the Brazilian Agricultural Research Corporation (Embrapa), which aims for biofortification in vitamin A precursors [9].

The crop also has potential for the production of edible seed oil. Its seed oil comprises about 70% unsaturated fatty acids, and it has a high content of monounsaturated fatty acids [10, 11, 12], so it is a good substitute for other lipid sources that have higher contents of saturated fatty acids. The oil is also rich in bioactive components such as vitamin E and carotenoids [13], which are important antioxidants in the human diet, in addition to protecting the oil itself against oxidative processes. In addition, this species is commonly cultivated in low-technology systems [14], making it fundamental to ensuring healthier diets and promoting food security in the regions where it is grown, particularly in less-developed regions and in the context of family-based farming.

Associated with its socio-economic importance, *C. moschata* germplasm commonly expresses high genetic variability in all regions where it occurs [15, 16, 17, 18], especially in Brazil [19, 20, 21]. Archaeological evidence indicates that this species was present in Latin America prior to colonisation, and appears to have already been an important component in the diet of the native peoples living there [22, 23, 24]. Currently, the variability of this vegetable in Brazil is closely tied to the human populations involved in its cultivation, who are predominantly family-based farmers. The selection practised over time by these populations, associated with the exchange of seeds between them, and the natural occurrence of hybridisation in the germplasm of this species has contributed to its increased variability. The high variability in agronomic, nutritional and bioactive characteristics displayed by *C. moschata* and the intercrossability of *Cucurbita* species have enabled these characteristics to be transferred from *C. moschata* to other species of this genus [25, 26, 27, 28]. This is of strategic importance and may aid the worldwide cultivation of species of the *Cucurbita* genus.

The usefulness of plant germplasm conserved in banks depends on the amount and quality of information associated with it, such as genetic and phenotypic data, which highlights the importance of its proper evaluation. On the other hand, the high volume of germplasm and limitations in resources and area available for the establishment of field trials commonly make its assessment difficult. In view of this, the FAO's Second Global Action Plan for Plant Genetic Resources for Food and Agriculture sets out guidelines that provide greater efficiency in the conservation and use of plant germplasm [29]. This is essential information for the management and use of germplasm [30, 31, 32, 33]. Evaluation of the germplasm maintained in banks

makes it possible to estimate the magnitude of the genetic and statistical parameters of characteristics of interest, which can provide information on the nature of variability observed for these traits, in addition to elucidating which characteristics or groups of characteristics most contribute to germplasm variability. From this assessment, it is also possible to assess the association between the characteristics evaluated. Together, the information obtained from these assessments is essential for optimising the use and management of plant germplasm.

The UFV Vegetable Germplasm Bank (BGH-UFV) maintains more than 350 accessions of *C. moschata*, constituting one of the largest collections of this species in Brazil [34]. This bank continually carries out work on the characterisation and evaluation of this germplasm [35], which has allowed the sources of resistance to important phyto-pathogenic agents to be identified [36], and its production [21] and nutritional aspects of fruits and seed oil to be improved [10, 37]. The potential of this germplasm as a source of genes for the improvement of this crop, along with the possibility of elucidating the genetic mechanisms linked to important production parameters, justifies the continuation of studies on its assessment and use.

This study therefore aimed to: a) agro-morphologically assess some of the *C. moschata* accessions maintained by BGH-UFV, b) analyse the genetic relationships of these agro-morphological characteristics, and c) analyse their agro-morphological variability, with a view to identifying earlier-flowering genotypes, genotypes with high total levels of carotenoids in the fruit pulp, and those with high potential for seed and seed oil productivity.

## Materials and methods

### Origin of germplasm and preparation of seedlings

In this study, we assessed 95 genotypes, comprising 91 accessions of *C. moschata* maintained in the BGH-UFV, and four control genotypes (Fig 1). The controls comprised the commercial

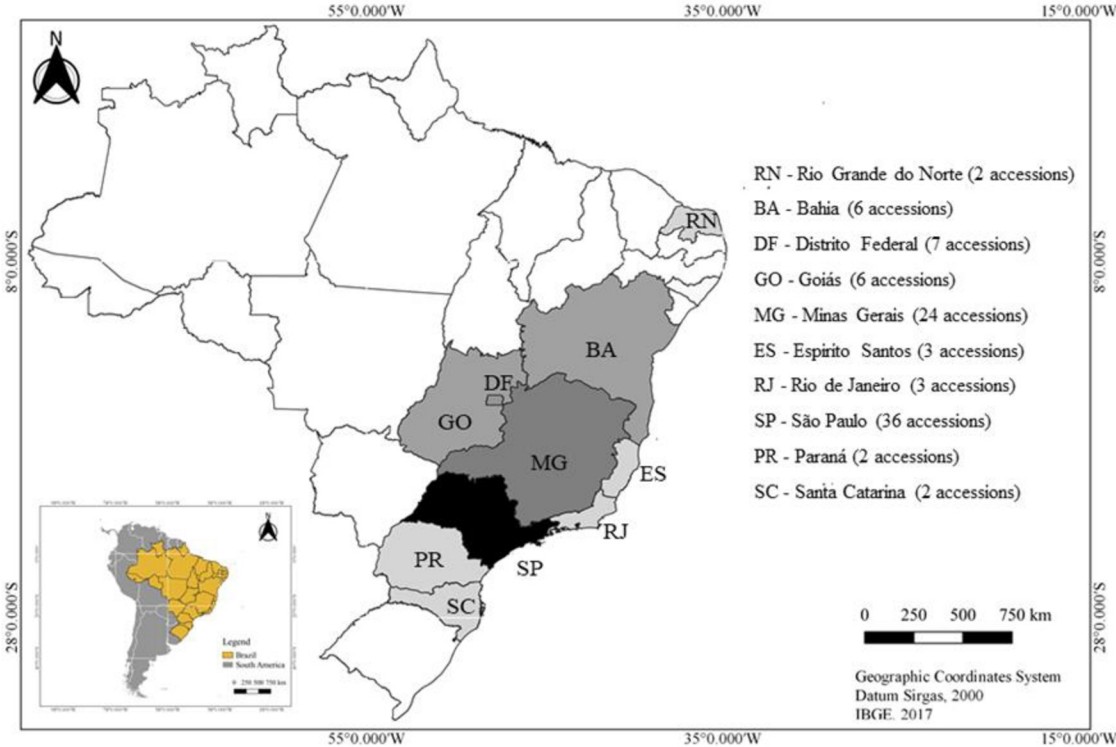

**Fig 1. Brazilian map showing the states of origin of the *C. moschata* accessions assessed in this study.**

hybrids Tetsukabuto and Jabras, and the cultivars Jacarezinho and Maranhão, all widely culti-vated and commercialised in Brazil. The accessions came from different regions of Brazil [35], and consisted, for the most part, of landraces collected from family-based farmers, who com-monly select the genotypes and conserve their seeds.

Seedlings were produced in a 72-cell expanded-polystyrene tray containing commercial substrate. Seedling transplantation and cultural treatments were carried out according to local recommendations for the cultivation of pumpkins [38].

## Experiment location and experimental design

The experiment was carried out from January to July 2016, at "Horta Velha" (200˚ 45'14" S, 420˚ 52'53" W and 648.74 m alt.), an experimental unit of the Agronomy Department of the Federal University of Viçosa, Viçosa-MG, Brazil.

The experiment was arranged in a Federer's augmented block design [39], with five replica-tions for each control. The four controls, also called common treatments, were randomly dis-tributed in each of the five blocks, and the 91 accessions, called regular treatments, were randomly assigned to all blocks. A spacing of 3x3 m between plants and rows was adopted, which resulted in a stand of 1,111 plants ha$^{-1}$. Each plot consisted of five plants, and all assess-ments were carried out from three central plants. The evaluations of fruit and seed characteris-tics were carried out on three fruits per plant.

## Assessments of agro-morphological aspects, total carotenoid content of fruit pulp, and seed and seed oil yields

For the assessment involving multi-categorical characteristics, we adopted the morphological descriptors suggested by Bioversity International and the European Cooperative Programme for Plant Genetic Resources (ECPPGR), plus some additional descriptors.

These descriptors comprised agro-morphological characteristics of plants, fruits, and seeds (S1 Table). Assessment was also based on agronomic characteristics, the total content of fruit pulp carotenoids, productivity of seeds, and seed oil productivity (Table 1).

The estimates of the total carotenoid (TC) and lutein contents (L) of fruit pulp were based on colorimetric parameters. For this, the fruit pulp colour was characterised with the aid of a manual tri-stimulus colorimeter, Colour Reader CR-10 Konica Minolta, by parameters related to luminosity, and the contribution of red (a) and yellow (b). The fruit pulp was characterised from a fruit from each of the three central plants of the plot. This was carried out on pulp from

**Table 1. Descriptors involving agronomic aspects of plants, fruits and seeds, used in the assessment of the *C. moschata* germplasm maintained by BGH-UFV.**

| Phase/organ | Descriptors |
| --- | --- |
| Reproductive phase | Accumulated degree-days for flowering (DDF). |
| Fruit | Number of fruits per plant (NFP), average mass of fruits (MF), productivity of fruits (PF), height of fruit (HF), diameter of fruit (DF), thickness of fruit peel (TFP), resistance of fruit peel to penetration (RFP), resistance of fruit pulp to penetration (RP), thickness of fruit pulp (PT), diameter of internal cavity of fruit (DIC), total content of fruit pulp carotenoids (TC), and the lutein content of fruit pulp (L). |
| Seed | Number of seeds per plant (NSF), mass of seeds per fruit (MSF), ratio of seed to fruit mass (MS/F), mass of one hundred seeds (MOH), productivity of seeds (PS), seed thickness (ST), seed length (SL), and seed width (SW). |
| Seed oil | Seed oil content (SOC) and seed oil productivity (SOP). |

four different parts of the fruit (part facing the sun, part facing the soil, part by the peduncle, and floral insertion part). The values of each parameter consisted of averages obtained from the pulp of fruits harvested from each of the plots' central plants. The estimates of TC were obtained using the equations proposed by [40], described below:

$$C = \sqrt{a^2 + b^2}$$

TC = 6.1226 + 1.7106 a

L = -6.3743 + 0.2818∗C

Where:

C corresponds to the saturation or chroma of the fruit pulp;

a corresponds to the contribution of red to the colour of fruit pulp;

b corresponds to the contribution of yellow to the colour of fruit pulp;

TC corresponds to the total content of fruit pulp carotenoids ($\mu$g g$^{-1}$ of fresh pulp mass); and

L corresponds to the lutein content of fruit pulp ($\mu$g g$^{-1}$ of fresh pulp mass).

The seed oil was extracted by cold pressing, with the aid of a 30- ton-capacity press, with the necessary adaptations for pressing. For this, the seeds were previously dried in a forced-air-circulation oven for 72 hours, at 23˚C. To standardise the process, 50 g seed samples were weighed from each accession and all samples were equally pressed for approximately 10 minutes.

## Estimation of genotypic values, components of variance and genetic-statistical parameters

Phenotypic data were analysed using restricted maximum likelihood (REML) procedures and the best linear unbiased prediction (BLUP). These procedures were carried out with the aid of the R program, using the "lme4" package [41]. The estimates of variance components were obtained from the REML procedure, while the genotypic values of accessions (BLUPS) and controls (BLUES) were obtained from the BLUP procedure. All estimates were based on the following model:

$$y = Wb + Xa + Zt + e$$

in which:

y corresponds to the phenotypic data vector;

b corresponds to the vector comprising the effect of blocks, assumed to be random;

a corresponds to the vector comprising the effect of accessions, assumed to be random:

t corresponds to the vector comprising the effect of controls, assumed to be fixed: and

e corresponds to the error vector.

The letters W, X and Z correspond to the incidence matrices of parameters b, a, and t, respectively, with the data vector y.

The estimates of variance components comprised the phenotypic ($\sigma^2_p$), genotypic ($\sigma^2_g$), and residual ($\sigma^2$) variances, and the variance associated with the block effect ($\sigma^2_b$). The genetic-

statistical parameters comprised the broad sense heritability ($h^2$), the selection accuracy ($A$), selection gain ($SG$), the phenotypic mean of the characteristics ($\mu$), and the genotypic ($CV_g$ %), phenotypic ($CV_P$ %), and residual ($CVr$ %) coefficients of variance. These were obtained from the following estimators: $h^2 = 1 - (Pev/2\sigma^2_g)$, where *Pev* corresponds to the prediction of error variance [42]: $A = \sqrt{1 - (Pev/\sigma^2_g)}$; $SG = h^2 * DS$, where *DS* corresponds to the selection differential, estimated from the average of the top 15% most promising accessions: $CV_g$ % = $(\sigma^2_g/\mu)$ x 100; $CV_P$ % = $(\sigma^2_p/\mu)$ x 100; e $CVr$ % = $(\sigma^2/\mu)$ x 100.

## Correlation analysis

This analysis was based on the matrix of genetic correlations, obtained from the following estimator:

$$rg = Cov\ (x,\ y)/\sqrt{\sigma^2_g(x)\ \sigma^2_g(y)}$$

in which;

*Cov* (x, y), corresponds to the genetic covariance between two variables X and Y, and $\sigma^2_g$ (x) and $\sigma^2_g$ (y) correspond to the genetic variances of variables X and Y, respectively.

The correlations were analysed using a procedure known as a *correlation network*, which allows all relationships between the variables under study to be analysed in relation to a specific function. This procedure also allows the direction and magnitude of the correlations to be distinguished. The direction is denoted by colours: dark green is used for the lines that connect positively-correlated variables, and red for the lines that connect negatively-correlated variables. The magnitude of the correlations is denoted by the thickness of the lines connecting the variables: the thicker the line, the greater the correlation. The significance of the correlations was analysed using Mantel's Z test at 1 and 5% probability. The correlation analysis was performed with the aid of the Genes program [43].

## Analysis of variability and clustering

The analysis of variability was carried out using both quantitative and multi-categorical information. For quantitative data, the distance matrix between the genotypes was obtained from the BLUPS estimates in the case of accessions, and from the BLUES in the case of the controls; the genetic distances were obtained based on the negative average Euclidean distance, with data standardisation.

The matrix was obtained from *negDistMat*, a function of the APCluster package [44] implemented in the R program, version 3.5.1 [45]. The distances d (x; y) between the accession pairs, exemplified here as any two accessions x $(x_1, \ldots, x_n)$ and y $(y_1, \ldots, y_n)$, were estimated from the following equation:

$$d(x, y) = -(1/v)\sqrt{\sum_{i=1}^{n} (x_i - y_i)^2}$$

in which v corresponds to the number of quantitative descriptors evaluated.

The distance matrix for the qualitative data was obtained using the arithmetic complement of the simple coincidence index. The variability analysis was performed from a single distance matrix, obtained from the sum of the distance matrices of the quantitative and qualitative data. For the sum of matrices, they were standardised and each received an equal weight in the sum procedure. The variability analysis was performed using the procedure known as the *Affinity*

*propagation* method [46]. The grouping was carried out from 100 independent rounds, aiming to assess the consistency of grouping.

The operation of *Affinity* initially involves the identification, in a set of components, of samples that will function as centres of this set. This method simultaneously considers all the set components as potential centres, i.e. as nodes in an interconnected network. Following the identification of potential centres, messages are transmitted between the set components along the network until a good set of centres and their corresponding groups emerge. The messages exchanged between the components in *Affinity* can be "responsiveness" $r$ $(i, k)$ and "availability $a$ $(i, k)$. This first case reflects the accumulated evidence of how appropriate point $k$ is to serve as an example for point $i$, considering all other potential examples for this point. The "availability", in turn, reflects the accumulated evidence of how appropriate it would be for point $i$ to choose point $k$ as an exemplar, considering the other points for which point $k$ can be an exemplar [46]. In the analysis of the present study, availability was initially established as zero.

A principal component analysis was implemented in order to identify the contribution of traits in the clustering of the genotypes. This analysis considered the data of quantitative and multi-categorical traits, according to the methodology of [47]; and was implemented using the FactoMineR package [48].

## Identification of promising accession groups and *per se* identification of accessions

In order to facilitate the identification of promising groups of accessions for each characteristic, we carried out a grouping of means of the genotypic values corresponding to the groups obtained from the analysis of variability. This was based on Tocher's method of grouping means. The identification *per se* of the most promising accessions for each trait was carried out by ranking the respective genotypic effects, genetic gain and the new predicted average of the accessions, and the top 15% were considered the most promising accessions.

## Results

### Variance components and genetic-statistical parameters of the agronomic aspects, total content of fruit pulp carotenoids, and the characteristics of seeds and seed oil

Estimates of the variance components and the genetic-statistical parameters are presented in Table 2. The estimates of genotypic variance were highest for number of seeds per fruit (NSF) and mass of seeds per fruit (MSF), decreasing to accumulated degree-days for flowering (DDF), and total content of fruit pulp carotenoids (TC). Among these variance estimates, only the genotypic variance of DDF was not significant. The estimates of variance associated with the block effect were low for all characteristics (Table 2).

For mass of seeds per fruit (MSF), number of seeds per fruit (NSF), total content of fruit pulp carotenoids (TC), and accumulated degree-days for flowering (DDF), most of the phenotypic variance was attributable to genotypic variance, with residual variance contributing less for most of the characteristics (Table 2).

As can also be seen in Table 2, most of the characteristics had high values for selection accuracy ($A$). Heritability estimates were 0.525, 0.495, and 0.774 for accumulated degree-days for accumulated-days for flowering (DDF), productivity of fruits (PF), and total content of fruit pulp carotenoids (TC), respectively, While productivity of seeds (PS) had a heritability of 0.481 and seed oil productivity (SOP) 0.291. Heritability was high (>0.50) for most of the characteristics, and very high for seed characteristics, such as mass of seeds per fruit (MSF), ratio of seed to

**Table 2. Estimates of variance components and genetic-statistical parameters of agronomic aspects, total content of fruit pulp carotenoids, and yields of seeds and seed oil.**

| Vegetative trait | | | | | | | | | | | |
|---|---|---|---|---|---|---|---|---|---|---|---|
| Traits | $\sigma_p$ | $\sigma_g$ | $\sigma$ | $\sigma_b$ | $A$ | $h^2$ | $SG$ | $Range$ | $\mu$ | $CV_g$ % | $CV_P$ % | $CVr$ % |
| DDF | 10781.493 | 6385.892[ns] | 3909.203 | 486.397800 | 0.725 | 0.525 | -92.947 | 120.0–820.4 | 606.642 | 13.172 | 17.116 | 10.306 |
| **Fruit traits** | | | | | | | | | | | |
| Traits | $\sigma_p$ | $\sigma_g$ | $\sigma$ | $\sigma_b$ | $A$ | $h^2$ | $SG$ | $Range$ | $\mu$ | $CV_g$ % | $CV_P$ % | $CVr$ % |
| NFP | 8.724 | 3.583[ns] | 4.614 | 0.527 | 0.655 | 0.429 | 2.303 | 1–15 | 4.783 | 39.575 | 61.752 | 44.909 |
| MF | 2.738 | 2.373** | 0.364 | 0.000 | 0.841 | 0.707 | 2.189 | 0.45–10.0 | 2.735 | 56.323 | 60.500 | 22.059 |
| PF | 73.954 | 38.598* | 29.279 | 6.076 | 0.704 | 0.495 | 7.817 | 0.7–44.6 | 12.946 | 47.989 | 66.427 | 41.796 |
| TC | 387.206 | 362.902** | 24.303 | 0.000 | 0.880 | 0.774 | 20.426 | 43.4–187.2 | 65.763 | 28.967 | 29.921 | 7.496 |
| **Seed and oil traits** | | | | | | | | | | | |
| Traits | $\sigma_p$ | $\sigma_g$ | $\sigma$ | $\sigma_b$ | $A$ | $h^2$ | $SG$ | $Range$ | $\mu$ | $CV_g$ % | $CV_P$ % | $CVr$ % |
| NSF | 25274.617 | 20784.317** | 2817.703 | 1672.597 | 0.844 | 0.712 | 167.873 | 78.6–805.7 | 454.188 | 31.741 | 35.003 | 11.685 |
| MSF | 490.881 | 465.357** | 16.141 | 9.382 | 0.899 | 0.808 | 27.428 | 4.4–119.3 | 51.929 | 41.541 | 42.665 | 7.736 |
| MS/F | 0.000142523 | 0.000125343** | 0.000015091 | 0.000002089 | 0.854 | 0.729 | 0.015 | 0.00–0.05 | 0.023 | 48.676 | 51.905 | 16.890 |
| MOHS | 7.395 | 4.391* | 3.003 | 0.000 | 0.721 | 0.519 | 2.210 | 6.3–23.6 | 11.701 | 17.908 | 23.240 | 14.809 |
| PS | 0.042 | 0.019[ns] | 0.016 | 0.006 | 0.694 | 0.481 | 0.187 | 0.01–0.9 | 0.269 | 51.241 | 76.185 | 47.022 |
| SOC | 13.010 | 0.462[ns] | 11.822 | 0.725 | 0.512 | 0.262 | 1.254 | 28.50–54.4 | 18.516 | 3.670 | 19.480 | 18.569 |
| SOP | 0.001743 | 0.000172[ns] | 0.001300 | 0.000 | 0.540 | 0.291 | 0.072 | 0.004–0.40 | 0.050 | 26.037 | 83.498 | 72.111 |

Accumulated degree-days for flowering (DDF), number of fruits per plant (NFP), average mass of fruit (MF), productivity of fruit (PF), total content of fruit pulp carotenoids (TC), number of seeds per fruit (NSP), mass of seeds per fruit (MSF), ratio of seed to fruit mass (MS/F), mass of one hundred seeds (MOHS), productivity of seeds (PS), seed oil content (SOC), and seed oil productivity (SOP). Components of variance involving phenotypic ($\sigma_p$), genotypic ($\sigma_g$), and residual ($\sigma$) variances, and the variance associated with the block effect ($\sigma_b$). Genetic-statistical parameters involving accuracy ($A$), broad-sense heritability ($h^2$), selection gain ($SG$), average ($\mu$), coefficients of genotypic ($CV_g$ %), phenotypic ($CV_P$ %), and residual variation ($CVr$ %). *ns* not significant; **, * significant at $p < 0.01$ and 0.05, respectively by the likelihood ration test.

fruit mass (MS/F), and number of seeds per fruit (NSF), and fruit characteristics, such as total content of fruit pulp carotenoids (TC) and average mass of fruit (MF), as shown in Table 2.

The high estimates of genotypic variance and heritability showed that considerable selection gain could be obtained for most of the characteristics (Table 2). For the number of accumulated degree-days for flowering (DDF), the gain was -92.947. It was also possible to obtain gains of 7.817 t ha⁻¹ for productivity of fruits (PF) and 20.426 μg g⁻¹ of fresh pulp mass for total content of fruit pulp carotenoids (TC), while the potential gains for productivity of seeds (PS) and seed oil productivity (SOP) were 0.187 and 0.072 t ha⁻¹, respectively (Table 2).

The phenotypic range between accessions for accumulated degree-days for flowering (DDF) was 120.0 to 820.4 (average 606. 642) (Table 2). The range for productivity of fruits (PF) was 0.7 to 44.6 t ha⁻¹ (average 12.946 t ha⁻¹), and that for total content of fruit pulp carotenoids (TC) was 43.4 to 187.2 μg g⁻¹ of fresh pulp mass (average 65.763 μg g⁻¹), while that for productivity of seeds (PS) was 0.01 to 0.9 t ha⁻¹ (average 0.269 t ha⁻¹). The phenotypic range between accessions for seed oil productivity (SOP) was 0.004 to 0.40 t ha⁻¹ (average 0.050 t ha⁻¹) (Table 2).

The greatest ranges between accessions for the coefficients of genotypic variation ($CV_g$%) were for mass of fruit (MF) and seed oil content (SOC), while for the coefficient of phenotypic variation ($CV_P$%), the greatest ranges between accessions were for seed oil productivity (SOP) and accumulated degree-days for flowering (DDF). The estimates of residual variation

coefficient ranged from 7.502 to 71.582 for total content of fruit pulp carotenoids (TC) and SOP, respectively (Table 2).

## Genotypic correlations

A genotypic correlation network analysis and visualisation of agronomic aspects, including the total content of fruit pulp carotenoids, and characteristics of seeds and seed oil is given in Fig 2, which shows cohesion of groups involving some of the fruit characteristics and those involving some of the characteristics of seeds. Cohesion is also shown between fruit productivity (PF) and other characteristics of this group, such as average mass of fruits (MF), diameter of internal cavity of fruit (DIC), height of fruit (HF), diameter of fruit (DF,), and thickness of fruit peel (TFP). As can be inferred from the colour and thickness of the lines, this set of variables showed high positive correlations. The highest correlations in this group were for PF with MF, and PF with DIC, with values equivalent to 0.61 and 0.54, respectively, both of which were significant ($p<0.01$). The productivity of fruits (PF) and number of fruits per plant (NFP) showed a correlation of 0.39, and each of these showed high correlations with the productivity of seeds (PS), 0.74 and 0.51, respectively all of which were significant ($p<0.001$), (Fig 2).

Accumulated degree-days for flowering (DDF) had low correlation with others characteristics. Seed oil content (SOC) had negative and low-magnitude correlations with soluble solids of fruit pulp (SS) and resistance of fruit pulp to penetration (RP) (Fig 2).

There was cohesion between the group of variables involved in seed productivity and variables such as the ratio of seed to fruit mass (MS/F), number of seeds per fruit (NSF), and mass of seeds per fruit (MSF). This set of variables had positive and high-magnitude correlations, of which the correlation of seed productivity (SP) with MS/F, equivalent to 0.56 and significant ($p<0.01$), was the highest. The group involving the mass of one hundred seeds (MOHS) and characteristics such as seed width (SW), seed thickness (ST), and seed length (SL) was also a cohesive group. This group had positive correlations, of which the correlation of MOHS with seed width SW, equivalent to 0.62 and significant ($p<0.01$), was the highest (Fig 2).

## Genetic variability and clustering

Cluster analysis, based on the agro-morphological aspects, the total content of fruit pulp carotenoids, and the characteristics related to the yields of seed and seed oil of the germplasm, placed the accessions into 16 groups (Table 3).

Based on the clustering pattern, high variability was observed between the accessions. About 17% of the genotypes were in group 11, together with the control, Jabras. Group 1, the second largest, contained 13.18% of the accessions and two controls, Jacarezinho and Maranhão. Groups 5 and 14 contained 10 and 11 accessions, respectively, making them the next largest groups formed. The grouping of genotypes in the other groups did not occur equitably and some of them contained only one genotype (Table 3).

The visual pattern of the clustering in heatmap format showed low similarity between the groups formed, as denoted by the predominance of yellow and orange colouring (Fig 3). Visual analysis of this clustering also shows homogeneity of the distances between groups, denoted by the uniformity of the colouring. The morphological pattern of fruits representative of part of the groups obtained with genotypes clustering is shown in the Fig 4.

The result of the principal components analysis (PCA) refers to the first 15 independent components, which explained 55.56% of the total variation observed between the genotypes (Fig 5). Component 1, which explained 7.35% of the total variation, had a greater contribution from quantitative variables, mainly from the average mass of fruits (MF), diameter of fruits (DF), and diameter of internal cavity of fruit (DIC). Component 2 had a greater contribution

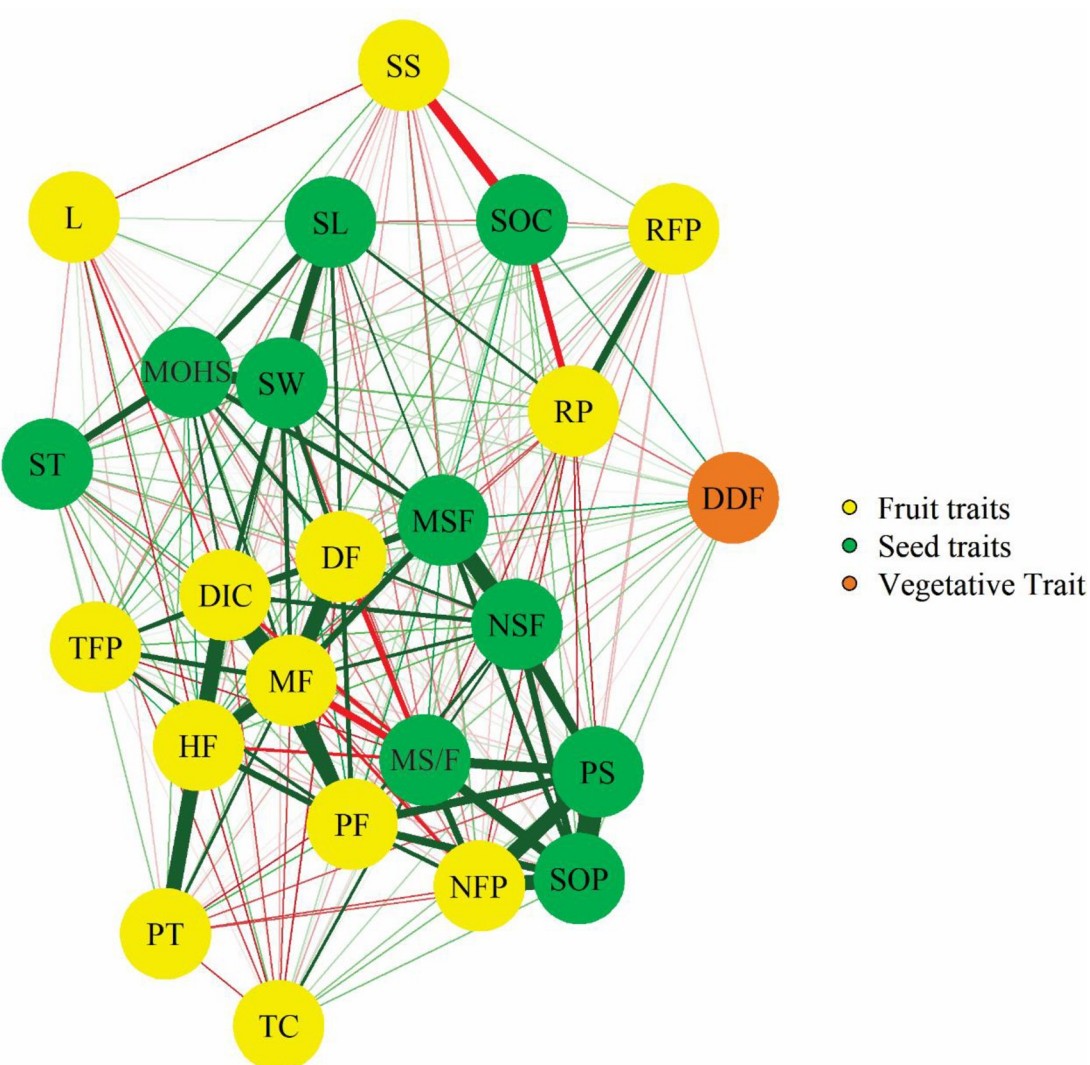

**Fig 2. Network of genotypic correlations of agronomic aspects, the total content of fruit pulp carotenoids, and the characteristics of seeds and seed oil of the *C. moschata* germplasm assessed in this study and maintained by the BGH-UFV.** The green and red lines denote positive and negative correlations, respectively. Thicker lines indicate greater magnitudes of correlation while the thinner lines indicate lesser magnitudes. Accumulated degree-days accumulated for flowering (DDF), number of fruits per plant (NFP), average mass of fruits (MF), productivity of fruits (PF), height of fruit (HF), diameter of fruit (DF), thickness of fruit peel (TFP), resistance of fruit peel to penetration(RFP), resistance of fruit pulp to penetration (RP), pulp thickness (PT), diameter of internal cavity of fruit (DIC), soluble solids of fruit pulp (SS), total content of fruit pulp carotenoids (TC), lutein content of fruit pulp (L), mass of seeds per fruit (MSF), productivity of seeds (PS), ratio of seed to fruit mass (MS/F), mass of one hundred seeds (MOHS), seed oil content (SOC), and seed oil productivity (SOP).

from the multi-categorical traits and explained 5.93% of the total variation (Fig 5). The result of PCA regarding the fifteen principal components and the relative contribution of traits in each component is provided in the S2 Table.

## Identification of promising clusters and *per se* identification of promising genotypes

In order to facilitate the visualisation of clusters with the most desirable characteristics, a grouping of means of clusters was performed by the Tocher method (Table 4).

**Table 3. Clustering of the *C. moschata* germplasm assessed in this study and maintained by BGH-UFV, based on agro-morphological aspects, the total content of fruit pulp carotenoids, and the yields of seeds and seed oil.**

| Clusters | Accessions |
|---|---|
| 1 | BGH-117(BA); BGH-5616A(DF); BGH-5630A(DF); BGH-6590(GO); BGH-4281(MG), BGH-4454A (MG), BGH-6116(MG); BGH-5472A(SP), BGH-5541(SP), BGH-5556A(SP); Jacarezinho(BR); Maranhão (BR). |
| 2 | BGH-4459A(MG); BGH-5548A(SP). |
| 3 | BGH-4590A(MG). |
| 4 | BGH-5653(BA); BGH-1927(MG), BGH-4681A(MG). |
| 5 | BGH-1749(BA); BGH-7219A(PR), BGH-7668(PR); BGH-5051(RJ); BGH-5453A(SP), BGH-5473A(SP), BGH-5544A(SP), BGH-5591A(SP), BGH-5593(SP), BGH-5596A(SP). |
| 6 | BGH-4610A(MG), BGH-5361A(MG); BGH-3333A(RJ); BGH-5440A(SP), BGH-5485A(SP). |
| 7 | BGH-5455A(SP), BGH-5598A(SP). |
| 8 | BGH-5624A(DF); BGH-6587A(GO), BGH-6595(GO); BGH-5247A(MG), BGH-6115(MG); BGH-5493A (SP) BGH-5494A(SP), BGH-5559A(SP). |
| 9 | BGH-315(DF); BGH-6593(GO); BGH-1004(MG); BGH-5499A(SP), BGH-5530A(SP), BGH-5606A(SP). |
| 10 | BGH-1961(ES); BGH-4516(MG), BGH-5248(MG), BGH-5648(MG), BGH-5659A(MG); BGH-5442(SP), BGH-5538(SP), BGH-5554A(SP). |
| 11 | BGH-95(BA); BGH-5638(DF); BGH-1945A(ES); BGH-6794(GO); BGH-4453(MG), BGH-4607A(MG), BGH-6155(MG); BGH-5301(SP), BGH-5451(SP), BGH-5528(SP), BGH-5551(SP), BGH-5552(SP), BGH-5553(SP), BGH-5560A(SP), BGH-5597(SP); Jabras(BR). |
| 12 | BGH-5649A(BA). |
| 13 | GBH-5694(DF); BGH-6099(RN); BGH-900(SP). |
| 14 | BGH-5240(BA); BGH-5639(DF); BGH-4287A(MG), BGH-4598A(MG), BGH-5224A(MG), BGH-6117A (MG); BGH-1461A(SC), BGH-6749(SC); BGH-5466(SP), BGH-5497(SP), BGH-5603(SP). |
| 15 | BGH-1992(ES); BGH-6594(GO); BGH-305A(MG); BGH-6096(RN); BGH-291(RJ); BGH-5456A(SP). |
| 16 | Tetsukabuto(BR). |

The letters next to the names refer to the initials of the genotypes' states of origin. Bahia (BA), Distrito Federal (DF), Espírito Santo (ES), Goiás (GO), Minas Gerais (MG), Paraná (PR), Rio Grande do Norte (RN), Rio de Janeiro (RJ), São Paulo (SP), and Santa Catarina (SC), Brazil (BR).

The lowest mean for accumulated degree-days for flowering (DDF) occurred in Group 16, which contained only the control Tetsukabuto, although most groups expressed intermediate averages for this characteristic (Table 4). The group with the highest mean for productivity of fruits (PF) was Group 4, formed by the accessions BGH-1927, BGH-4681A, and BGH-5653. This group also expressed one of the highest averages for mass of fruits (MF) and an intermediate average for number of fruits per plant (NFP). As for the total content of fruit pulp carotenoids (TC), the highest average occurred in Group 7, formed by the accessions BGH-5455A and BGH-5598A. Groups 1 and 6 expressed the highest averages for seed (PS) and seed oil productivity (SOP). Group 1 contained the largest number of accessions (Table 4).

The identification *per se* of the most promising accessions for each trait, based on their respective genotypic effects, is shown in Tables 5 and 6. Also in these tables are the estimates, for each accession, of their genetic gains and the new predicted average for each trait.

The selected accessions had averages for accumulated degree-days for flowering (DDF) that were much lower than the general average of the accessions (606.64) and the average of the controls (526.41), with their new predicted averages ranging from 474.39 to 251.09, and genetic gains from -132.25 to -355.55. Notably, the accessions BGH-6749, BGH-5639, and BGH-2191 were the most promising for DDF (Table 5).

For productivity of fruits (PF), the selected accessions had higher averages than the general average of the accessions (12.95 t ha$^{-1}$) and the average of the controls (11.85 t ha$^{-1}$), with their

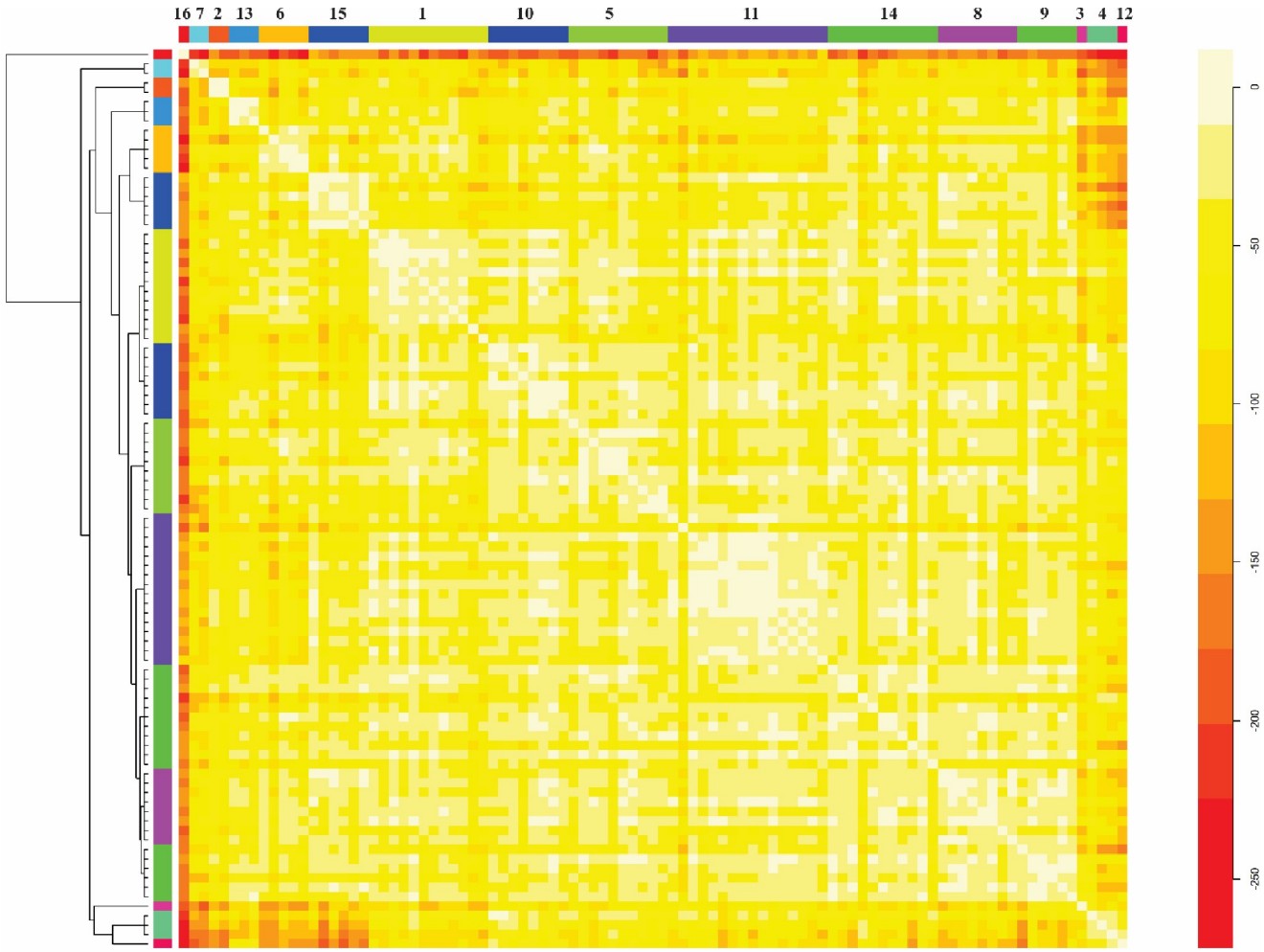

**Fig 3. Heatmap and hierarchical clustering of the genetic distances of the *C. moschata* accessions, based on agro-morphological traits, the total content of fruit pulp carotenoids, and the yields of seeds and seed oil.** The coloured bars on the upper and lower axis correspond to the groups obtained in the clustering. The dissimilarity between each pair of accessions and between groups is indicated by the colour, which varies from white to red. Red indicates the pairs of genotypes with the highest dissimilarity and white indicates the pairs of genotypes with lowest dissimilarities.

new predicted averages ranging from 15.49 to 29.27 t ha$^{-1}$. As for total content of fruit pulp carotenoids (TC), the selected accessions also had much higher averages than the general average of the accessions (65.76 μg g$^{-1}$ of fresh weight) and that of the controls (65.58 μg g$^{-1}$ of fresh weight). The new averages predicted for this characteristic among those selected ranged from 72.34 to 179.46 μg g$^{-1}$ of fresh pulp mass, and the most promising accessions for this characteristic were BGH-5455A and BGH-5598A (Table 5).

The identification *per se* of the most promising accessions for productivity of seeds (PS), seed oil content (SOC) and seed oil productivity (SOP), together with their respective genetic gains and new predicted averages for these characteristics is shown in Table 6.

As for productivity of seeds (PS), the new predicted averages among the selected accessions ranged from 0.33 to 0.58 t ha$^{-1}$ and the genetic gains from 0.06 to 0.31 t ha$^{-1}$. Notably, the accessions BGH-4610A, BGH-5485A, and BGH-6590 were the most promising for this characteristic (Table 6). The selected accessions displayed small differences in seed oil content (SOC); however, the average of these was higher than that of the controls (16.73%). Finally, for seed

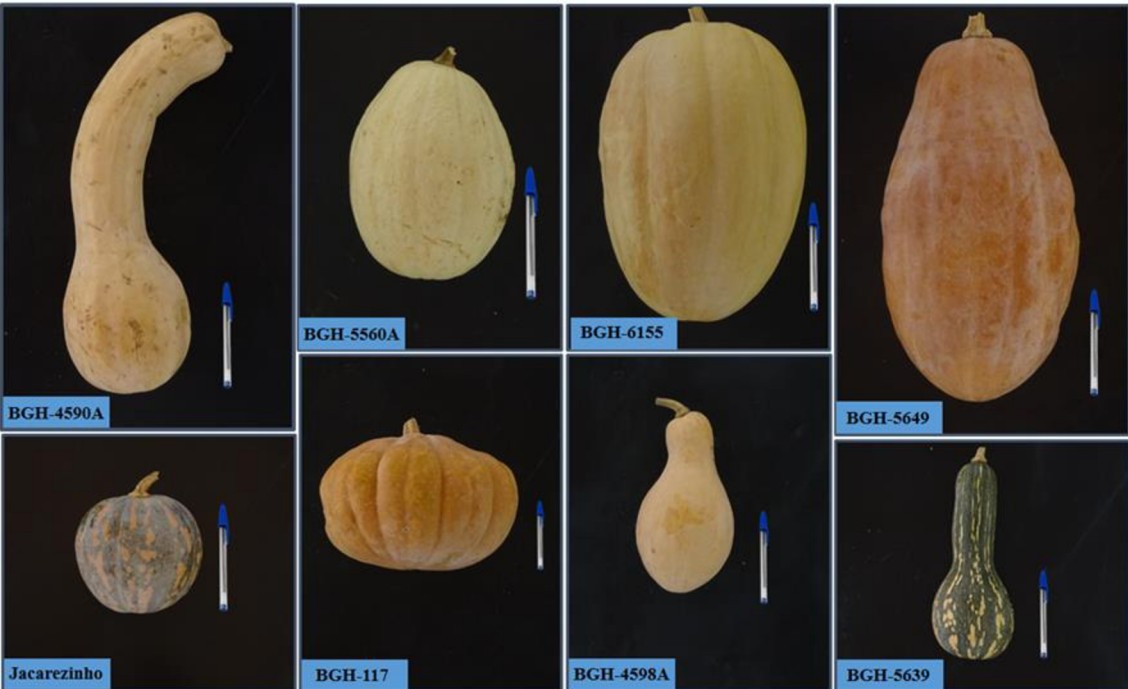

**Fig 4. Figure showing the morphological pattern of fruits representative of part of the groups obtained with genotypes clustering.**
BGH-4590A (group 3), BGH-5560A and BGH-6155 (group 11), BGH-5649A (group 12), Jacarezinho and BGH-117 (group 1), BGH-4598A and BGH-5639 (group 14), BGH-5548A (group 2), BGH-5453A and BGH-5544A (group 5), and BGH-900 (group 13).

oil productivity (SOP), the new predicted averages ranged from 0.12 to 0.13 t ha$^{-1}$ and the genetic gains from -0.07 to -0.08 t ha$^{-1}$. The accessions BGH-5485A, BGH-4610A, and BGH-5472A were the most promising for this characteristic (Table 6).

## Discussion

### Variance components and genetic-statistical parameters of the agronomic aspects, total content of fruit pulp carotenoids, and the characteristics of seeds and seed oil

As with other species, the usefulness of *C. moschata* germplasm conserved in banks depends on the level and quality of information associated with it [30, 31, 32, 33, 49]. The samples of *C. moschata* maintained by BGH-UFV constitute one of the largest collections of this species in Brazil [34]. Studies involving the assessment of this germplasm have allowed the identification of accessions with crucial characteristics for this crop, such as phytopathogenic resistance, and for its genetic improvement in terms of production and nutritional aspects of its fruits and seed oil [10, 21, 36, 37]. Although BGH-UFV maintains more than 350 accessions of *Cucurbita* ssp. [35], part of this germplasm has not yet been assessed, demonstrating the importance of continuing these studies.

Most of the *C. moschata* germplasm express vigorous growth and indeterminate growth habit [50], and *C. moschata* plants commonly occupy a large area of cultivated land, making it difficult to phenotypically assess its germplasm in experimental designs such as in randomised blocks. The main limitation in the evaluation of *C. moschata* germplasm in randomised blocks is the difficulty of ensuring satisfactory homogeneity throughout the experimental area. In

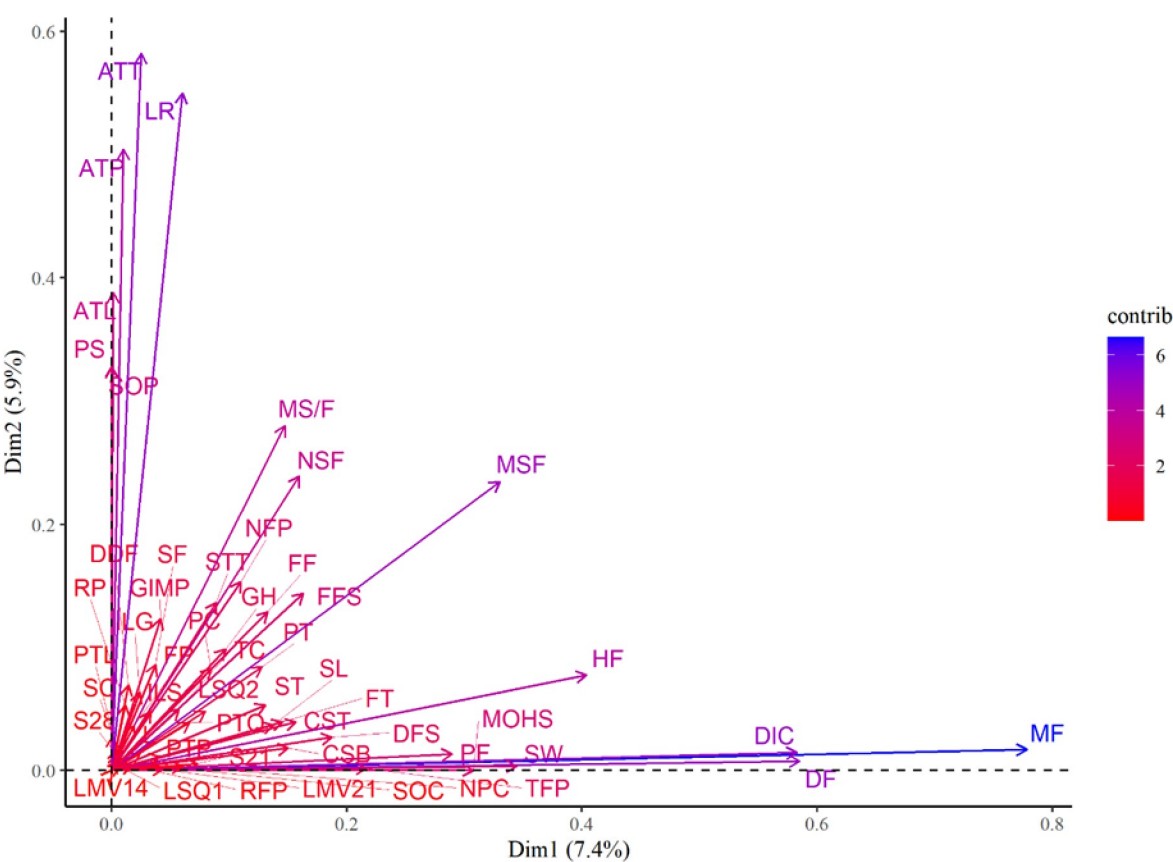

**Fig 5. Dispersion of quantitative and multi-categorical characteristics in relation to the first two components:** Leaf SPAD index at 21 days after transplanting (S21) and at 28 days (S28); Length of Main Vine at 14 days after transplanting (LMV14) and at 21 (LMV21); accumulated Degree-Days for Flowering (DDF); Number of Fruits per Plant (NFP); average Mass of Fruits (MF); Productivity of Fruits (PF); Height of Fruit (HF); Diameter of Fruit (DF); Thickness of Fruit Peel (TFP); Resistance of Fruit peel to Penetration (RFP); Resistance of fruit pulp to Penetration (RP); thickness of fruit pulp (PT); Diameter of Internal Cavity of fruit (DIC); Total Content of fruit pulp carotenoids (TC), and Lutein (L); Number of Seeds per plant (NSF); Mass of Seeds per Fruit (MSF); ratio of seed to fruit mass (MS/F); Mass of one Hundred Seeds (MOH); Productivity of Seeds (PS); Seed Thickness (ST); Seed Length (SL); Seed Width (SW); Seed Oil Content (SOC); Seed Oil Productivity (SOP). Growth habit (GH); stem colour (SC); intensity of leaf green (ILG), leaf silvering (LS), intensity of leaf silvering (ILS); leaf serration (LS); presence of trichomes in the leaves (PAL); amount of trichomes in the adaxial surface of leaves (ATT); amount of trichomes in the abaxial surface of leaves (ATL); leaf recess (LR); presence of trichomes in the petiole (PTP); amount of trichomes in the petiole (ATP); green intensity of male pedicel (GIMP); format of fruits (FF); format of peduncle (FP); number of colours of fruit peel (NPC); topography of fruit surface (FT); format of floral scar (FFS); peel texture (PT); predominant colour of fruit peel (PC); depth of fruits slices (DFS); seed format (SF); aspect of seed tegument (AST); seed tegument texture (STT); colour of seed tegument (CST); colour of seed border (CSB).

addition, the germplasm seed samples kept in banks in most cases are small, making it impossible to repeat accessions throughout the experimental area and assess quantitative characteristics. In view of this, we proposed in this study to evaluate part of the *C. moschata* germplasm maintained at BGH-UFV using the design known as Federer's augmented blocks [39]. The details of all aspects inherent to this design are very well described by Federer and, according to him, the design circumvents the limitations mentioned above and can be adopted even when the propagating material is insufficient for the establishment of more than one plot and where the quantity of samples to be evaluated is too great.

The present study describes the evaluation of one of the largest germplasm volumes of *C. moschata*. The high estimates of genotypic variance for characteristics related to seed production observed in this study corroborate those reported by [51], who also observed higher

**Table 4. Grouping of means of the genotypic values of the groups obtained in the analysis of variability for agro-morphological aspects, the total content of fruit pulp carotenoids, and productivities of seed and seed oil.**

| Groups | DDF | NFP | MF | PF | TC | NSF | MSF | MS/F | MOHS | PS | SOC | SOP |
|---|---|---|---|---|---|---|---|---|---|---|---|---|
| 1 | 19.15[b] | 0.48[b] | -0.22[d] | 1.46[c] | 2.09[b] | 94.22[b] | 0.52[b] | 0.52[b] | 0.63[c] | 0.12[a] | -0.07[a] | 0.04[a] |
| 2 | 32.20[b] | -0.69[b] | -0.58[e] | -2.80[c] | -8.77[b] | -95.99[b] | -0.80[b] | -0.80[c] | -1.54[d] | -0.08[b] | -0.16[a] | -0.03[b] |
| 3 | -35.38[b] | 0.15[b] | 1.17[b] | 5.61[b] | -7.46[b] | -207.22[c] | -1.22[b] | -1.22[d] | 3.62[a] | -0.05[b] | 0.12[a] | -0.02[b] |
| 4 | -19.47[b] | -0.67[b] | 4.53[a] | 10.41[a] | -4.95[b] | 129.02[b] | -0.53[b] | -0.53[c] | 1.66[b] | 0.02[b] | 0.07[a] | 0.01[b] |
| 5 | 2.76[b] | 0.33[b] | 0.40[c] | 2.89[c] | -8.54[b] | 44.57[b] | -0.40[b] | -0.40[c] | -0.85[c] | 0.02[b] | 0.01[a] | 0.01[b] |
| 6 | 11.73[b] | 2.69[a] | -1.23[g] | 1.50[c] | -0.84[b] | 79.53[b] | 1.76[b] | 1.76[a] | -1.16[d] | 0.21[a] | -0.01[a] | 0.07[a] |
| 7 | 36.35[b] | 1.24[a] | -1.36[g] | -2.48[c] | 111.02[a] | 13.62[b] | 1.90[b] | 1.90[a] | -0.08[c] | 0.06[b] | 0.06[a] | 0.03[b] |
| 8 | -9.88[b] | 0.14[b] | -0.47[d] | -0.09[c] | 2.01[b] | 43.54[b] | 0.56[b] | 0.56[b] | -1.79[d] | -0.01[b] | -0.14[a] | -0.01[b] |
| 9 | -8.08[b] | -0.66[b] | -0.93[f] | -3.83[c] | -6.11[b] | -36.84[b] | 0.42[b] | 0.42[b] | -0.41[c] | -0.06[b] | 0.02[a] | -0.02[b] |
| 10 | 40.41[b] | -0.65[b] | 0.84[b] | -0.10[c] | -5.74[b] | 151.09[b] | -0.3[b] | -0.31[c] | 0.97[b] | 0.00[b] | -0.03[a] | 0.00[b] |
| 11 | -10.76[b] | -0.80[b] | 0.29[c] | -1.32[c] | -1.03[b] | -96.72[b] | -0.79[b] | -0.79[c] | 0.91[b] | -0.09[b] | -0.05[a] | -0.03[b] |
| 12 | 127.82[a] | -1.17[b] | 4.33[a] | 5.60[b] | -11.10[b] | 228.21[a] | -0.65[a] | -0.65[c] | 2.62[a] | 0.03[b] | 0.20[a] | 0.02[b] |
| 13 | 82.43[a] | -0.30[b] | -0.24[d] | -1.24[c] | -5.32[b] | -46.40[b] | -0.02[b] | -0.02[c] | -0.19[c] | -0.03[b] | 0.11[a] | -0.01[b] |
| 14 | -66.85[b] | -0.17[b] | -0.02[d] | -0.47[c] | 1.13[b] | 22.01[b] | 0.48[b] | 0.48[b] | -0.08[c] | 0.01[b] | 0.04[a] | 0.00[b] |
| 15 | -43.65[b] | 0.72[b] | -1.39[g] | -2.61[c] | -0.99[b] | -243.74[c] | -0.86[b] | -0.86[c] | -1.34[d] | -0.08[b] | -0.03[a] | -0.03[b] |
| 16 | -138.72[c] | -0.06[b] | -1.32[g] | -4.92[c] | -10.69[b] | -393.92[d] | -1.82[c] | -1.82[d] | 0.07[c] | -0.22[c] | -5.50[b] | -0.05[c] |

The genotypic values of the accessions (BLUPS) and controls (BLUES) vary from negative to positive; therefore the signal of group means is a reflection of the genotypic values in each group. Accumulated degree-days accumulated for flowering (DDF), number of fruits per plant (NFP), average mass of fruits (MF), productivity of fruits (PF), total content of fruit pulp carotenoids (TC), number of seeds per fruit (NSF), mass of seeds per fruit (MSF), ratio of seed to fruit mass (MS/F), mass of one hundred seeds (MOHS), productivity of seeds (PS), seed oil content (SOC), and seed oil productivity (SOP). The letters a, b, c, d, f, and g refer to the groups formed in the clustering of means obtained by the Tocher method.

estimates of genotypic variance for the number of seeds per fruit and flowering characteristics, and also a greater contribution of genotypic variance to the phenotypic variance in these characteristics. Additionally, most of the characteristics assessed in this study gave high estimates of heritability (>0.50), considering the classification of [52], especially the characteristics of seeds such as mass of seeds per fruit (MSF), ratio of seed to fruit mass (MS/F), and number of seeds per fruit (NSF), as well the aspects related to fruits, such as total content of fruit pulp carotenoids (TC) and mass of fruit (MF). High estimates of heritability point to a greater correlation between the phenotype and the genotype [53], indicating that most of the variability observed for these characteristics resulted from genotypic effects.

The high estimates of genotypic variances may be associated with the quantitative nature of these characteristics, which may be the result of the influence of a high number of genes [54]. Most of the germplasm evaluated in this study came from the land of family-based farmers, who do not carry out selection either for seed characteristics or to obtain earlier-flowering genotypes. As already mentioned, the exchange of seeds between farmers and the natural occurrence of hybridisation between populations of *C. moschata* has increased the variability of this species, even for characteristics for which selection is commonly carried out, such as fruit productivity.

Considerable predicted gains were obtained for most of the characteristics, considering the overall average of accessions. This result was associated with the high estimates of genotypic variance and heritability observed for most of the characteristics (Table 2).

The average relationship between the coefficient of genetic variation and the residual coefficient was close to one unit for most of the characteristics. Although the estimates of the residual coefficients of variation for most characteristics were high, in general they tended to be lower in

**Table 5. Estimates of the genotypic effects, genetic gain and new predicted averages for the accumulated degree-days for flowering (DDF), fruit productivity (PF) and total content of fruit pulp carotenoids (TC), for the top 15% most promising accessions and the controls.**

| | DDF | | | | PF | | | | TC | | |
|---|---|---|---|---|---|---|---|---|---|---|---|
| Accessions | g | Gain | New Average | Accessions | g | Gain | New Average | Accessions | g | Gain | New Average |
| BGH-6749 | -291.35 | -355.55 | 251.09 | BGH-4453 | 17.32 | 16.32 | 29.27 | BGH-5455A | 113.85 | 113.70 | 179.46 |
| BGH-5639 | -152.11 | -216.29 | 390.35 | BGH-5653 | 5.60 | 15.44 | 28.38 | BGH-5598A | 108.19 | 108.03 | 173.80 |
| BGH-291 | -119.80 | -183.97 | 422.67 | BGH-5544A | 13.82 | 12.82 | 25.76 | BGH-1461A | 14.18 | 14.03 | 79.80 |
| BGH-5638 | -102.40 | -166.57 | 440.07 | BGH-4681A | 11.74 | 10.74 | 23.69 | BGH-5616A | 11.50 | 11.35 | 77.12 |
| BGH-6587A | -91.41 | -155.57 | 451.07 | BGH-5224A | 10.21 | 9.21 | 22.16 | BGH-6794 | 11.08 | 10.93 | 76.70 |
| BGH-5624A | -83.56 | -147.73 | 458.91 | BGH-6587A | -4.26 | 8.35 | 21.29 | BGH-5556A | 9.85 | 9.70 | 75.47 |
| BGH-1004 | -83.32 | -147.48 | 459.16 | BGH-4590A | 5.61 | 4.61 | 17.55 | BGH-5606A | 8.78 | 8.63 | 74.40 |
| BGH-1749 | -76.12 | -140.28 | 466.36 | BGH-5649 | 5.60 | 4.59 | 17.54 | BGH-5497 | 8.73 | 8.58 | 74.34 |
| BGH-5301 | -76.12 | -140.28 | 466.36 | BGH-5051 | 5.30 | 4.30 | 17.25 | BGH-5451 | 8.68 | 8.53 | 74.29 |
| BGH-5456A | -75.88 | -140.04 | 466.60 | BGH-5596A | 4.52 | 3.52 | 16.46 | BGH-5493A | 8.62 | 8.47 | 74.24 |
| BGH-5485A | -75.88 | -140.04 | 466.60 | BGH-5248 | 4.09 | 3.09 | 16.03 | BGH-5247A | 7.34 | 7.19 | 72.95 |
| BGH-5530A | -75.88 | -140.04 | 466.60 | BGH-5472A | 4.06 | 3.06 | 16.00 | BGH-6749 | 6.91 | 6.76 | 72.53 |
| BGH-6794 | -68.94 | -133.11 | 473.53 | BGH-5473A | 3.62 | 2.62 | 15.57 | BGH-6587A | 13.49 | 6.71 | 72.47 |
| BGH-4598A | -68.09 | -132.25 | 474.39 | BGH-5556A | 3.54 | 2.54 | 15.49 | BGH-95 | 6.73 | 6.58 | 72.34 |
| Average | | | 606.64 | Average | | | 12.95 | Average | | | 65.76 |
| Controls | | | | Controls | | | | Controls | | | |
| | BLUES | Gain | New Average | | BLUES | Gain | New Average | | BLUES | Gain | New Average |
| Jabras | -192.84 | -54.12 | 413.99 | Jabras | 8.53 | 8.87 | 20.88 | Jabras | 14.93 | 16.19 | 80.68 |
| Tetsukabuto | -138.72 | 0.00 | 468.11 | Tetsukabuto | -1.03 | -0.69 | 11.31 | Tetsukabuto | 0.23 | 1.49 | 65.98 |
| Maranhão | 3.99 | 142.71 | 610.82 | Maranhão | -4.60 | -4.26 | 7.75 | Maranhão | -5.13 | -3.88 | 60.61 |
| Jacarezinho | 5.89 | 144.61 | 612.72 | Jacarezinho | -4.92 | -4.57 | 7.43 | Jacarezinho | -10.69 | -9.43 | 55.06 |
| Average | | | 526.41 | Average | | | 11.85 | Average | | | 65.58 |

relation to their corresponding coefficients of genotypic variability, which demonstrates that most of the variability expressed by germplasm was due to genetic factors (Table 2).

## Genetic correlation network

Analysis of correlations between characteristics has been widely used in plant breeding, where often a high number of characteristics must be considered simultaneously [55, 56]. This analysis is often used to assist in indirect selection for certain characteristics [55, 57]. However, as highlighted by [58], in cases where one intends to practise indirect selection for a primary characteristic by means of a secondary one, the heritability of the latter characteristic must be greater than that of the former for efficient selection. In view of this, the selection of genotypes with higher average mass of fruits (MF) seems to be a promising alternative for obtaining higher fruit productivity in *C. moschata*.

It should, however, be highlighted that when selecting genotypes for increasing fruit productivity in *C. moschata*, crucial aspects for their acceptability in the consumer market, such as the shape and size of fruits, must be considered. Currently, important pumpkin consumption centres like the state of Minas Gerais and most of the southeast region of Brazil demand smaller fruits, and most of the consumption in these regions is represented by fruits from hybrid cultivars, such as Jabras and Tetsukabuto, which have a globular shape and weigh from 2 to 3 kg [14]. On the other hand, in the north and northeast regions of Brazil, larger fruits, which are commonly sold in slices, are more acceptable. The prevention of waste and the ease

**Table 6. Estimates of the genotypic effects, genetic gain and new predicted averages for the productivity of seeds (PS), seed oil content (SOC), and seed oil productivity (SOP), for the top 15% most promising accessions and the controls.**

| Accessions | PS | | | Accessions | SOC | | | Accessions | SOP | | |
|---|---|---|---|---|---|---|---|---|---|---|---|
| | g | Gain | New Average | | g | Gain | New Average | | g | Gain | New Average |
| BGH-4610A | 0.34 | 0.31 | 0.58 | BGH-7219A | 0.43 | -0.98 | 17.53 | BGH-5485A | 0.01 | -0.07 | 0.13 |
| BGH-5485A | 0.30 | 0.28 | 0.54 | BGH-5649 | 0.20 | -1.21 | 17.30 | BGH-4610A | 0.01 | -0.07 | 0.13 |
| BGH-6590 | 0.29 | 0.26 | 0.53 | BGH-5653 | 0.16 | -1.25 | 17.27 | BGH-5472A | 0.01 | -0.07 | 0.13 |
| BGH-5556A | 0.22 | 0.19 | 0.46 | BGH-5466 | 0.16 | -1.25 | 17.27 | BGH-5556A | 0.01 | -0.07 | 0.12 |
| BGH-5472A | 0.22 | 0.19 | 0.46 | BGH-900 | 0.16 | -1.26 | 17.26 | BGH-6590 | 0.01 | -0.07 | 0.12 |
| BGH-5544A | 0.19 | 0.17 | 0.44 | BGH-6155 | 0.16 | -1.26 | 17.26 | BGH-5544A | 0.01 | -0.07 | 0.12 |
| BGH-5440A | 0.18 | 0.15 | 0.42 | BGH-5544A | 0.15 | -1.26 | 17.25 | BGH-4281 | 0.01 | -0.08 | 0.12 |
| BGH-4281 | 0.16 | 0.13 | 0.40 | BGH-6794 | 0.15 | -1.26 | 17.25 | BGH-5440A | 0.01 | -0.08 | 0.12 |
| BGH-5361A | 0.16 | 0.13 | 0.40 | BGH-5472A | 0.15 | -1.27 | 17.25 | BGH-5630A | 0.01 | -0.08 | 0.12 |
| BGH-5630A | 0.15 | 0.12 | 0.39 | BGH-305A | 0.14 | -1.27 | 17.24 | BGH-5473A | 0.01 | -0.08 | 0.12 |
| BGH-5473A | 0.15 | 0.12 | 0.39 | BGH-5455A | 0.13 | -1.28 | 17.23 | BGH-5361A | 0.01 | -0.08 | 0.12 |
| BGH-5453A | 0.13 | 0.10 | 0.37 | BGH-5240 | 0.12 | -1.29 | 17.23 | BGH-5453A | 0.00 | -0.08 | 0.12 |
| BGH-4287A | 0.10 | 0.08 | 0.34 | BGH-4681A | 0.12 | -1.29 | 17.22 | BGH-5455A | 0.00 | -0.08 | 0.12 |
| BGH-4454A | 0.09 | 0.06 | 0.33 | BGH-4590A | 0.12 | -1.29 | 17.22 | BGH-5466 | 0.00 | -0.08 | 0.12 |
| Average | | | 0.27 | Average | | | 18.52 | Average | | | 0.11 |
| Controls | | | | Controls | | | | Controls | | | |
| | BLUES | Gain | New Average | | BLUES | Gain | New Average | | BLUES | Gain | New Average |
| Jacarezinho | 0.28 | 0.30 | 0.53 | Jacarezinho | 0.39 | 2.38 | 18.98 | Jacarezinho | 0.05 | 0.01 | 0.10 |
| Maranhão | 0.03 | 0.06 | 0.29 | Maranhão | -0.91 | 1.07 | 17.67 | Maranhão | 0.01 | -0.03 | 0.06 |
| Jabras | -0.14 | -0.12 | 0.11 | Jabras | -1.40 | 0.59 | 17.19 | Jabras | -0.03 | -0.08 | 0.01 |
| Tetsukabuto | -0.22 | -0.19 | 0.04 | Tetsukabuto | -5.50 | -3.52 | 13.08 | Tetsukabuto | -0.05 | -0.09 | 0.00 |
| Average | | | 0.24 | Average | | | 16.73 | Average | | | 0.04 |

of transport are determining aspects for the acceptability of fruit shapes, and the search for greater productivity in the cultivation of *C. moschata* must therefore also consider these characteristics, equating them with aspects such as the number of fruits per plant (NFP), height of fruit (HF) and diameter of fruit (DF).

Based on the correlations obtained in this study, the simultaneous consideration of aspects such as higher number of fruits per plant (NFP), higher productivity of fruits (PF) and higher ratio of seed to fruit mass (MS/F) seems to be a promising alternative for obtaining higher seed productivity (PS) in *C. moschata*. The heritability estimates obtained for these characteristics (>0.42), suggest that reasonable gains are feasible with selection for each one of them (Table 2). With this, besides greater PF and NFP, the selection of genotypes with higher PS should also prioritise greater translocation of photoassimilates for seed production, something indicated by a higher ratio of seed to fruit mass (MS/F).

Despite its applicability, correlation analysis has some limitations, and, as warned by [59], the quantification and interpretation of the correlation coefficients between two or more characteristics can result in errors during the selection process. According to them, this occurs because high estimates of correlations between these characteristics may be the effect of one or more secondary characteristics. It is therefore recommended that analysis of the association between a primary and secondary characteristic be accompanied by information on the direct and indirect effects of secondary variables on the primary [60], an approach currently known as path analysis [59].

Despite some limitations, correlation analysis has proven to be quite useful in plant breeding, mainly in the indirect selection for one or more main characteristics that have low heritability or are difficult to assess. This indirect selection is based on secondary characteristics with greater heritability or ease of assessment, providing faster genetic gains than with direct selection. In fact, correlation analysis has assisted in the indirect selection for characteristics of roots [61], for productivity in different crops [62, 63, 64], and for nutritional aspects and quality of fruits [65, 66]. Correlation analysis can also be very useful in the characterisation and management of plant germplasm, as it may optimise the choice and number of descriptors to be used in this process.

## Genetic variability and clustering

The analysis of variability provides important assistance in the initial phase of plant breeding programmes and in the management of plant germplasm. In this first case, it provides allocation of accessions in groups, guiding crossbreeding. *C. moschata* is allogamous, and analysing the variability of its germplasm can assist in the orientation of crossings between more diverse genotypes, thereby aiding the exploration of hybrid vigour [67, 68]. Variability analysis also allows duplicates in the germplasm collections [69, 70, 71], which correspond to pairs or groups of accessions with high similarity, to be identified. In fact, it is estimated that less than 30% of the accessions maintained in the collections worldwide are distinct, which hinders their maintenance [29]. Therefore, in addition to optimising the use of germplasm, variability analysis reduces the cost of its maintenance by reducing its volume [72].

The accessions of *C. moschata* assessed in this study displayed high genetic variability in their agro-morphological characteristics, the total content of fruit pulp carotenoids (TC), and the productivity of seeds (PS) and seed oil (SOP), resulting in the formation of 16 clusters (Table 3). The clustering of Jacarezinho and Maranhão in the same group (Group 1) reflects its consistency since these two cultivars have similar characteristics.

Clustering did not reflect a smaller genetic distance between those accessions from the same state or geographic region of Brazil. Group 11, for example, grouped accessions from different states and regions; and the preponderance of accessions from Minas Gerais (MG) and São Paulo (SP) in this group was probably only a result of the greater number of accessions from these states. This trend was repeated for other groups with higher numbers of accessions such as 1, 5 and 14. A study involving the assessment of *C. moschata* accessions from different regions of Brazil and maintained at BGH-UFV [73] also did not report smaller genetic distance between the accessions from the same state or region.

It is notable that the two hybrids used as controls, Jabras and Tetsukabuto, clustered in different groups. Although they have similar fruit shape and size, the groups to which they were allocated differed in most characteristics (Table 4), and their different genotypic values for most characteristics (Tables 5 and 6) justified their clustering in different groups. Tetsukabuto, which is an interspecific hybrid between *C. moschata* and *C. maxima* [74], corresponded to the group with lowest genotypic average for accumulated degree-days for flowering (DDF), in addition to expressing genotypic averages quite different from the other groups in relation to the characteristics of seeds and seed oil (Table 4), justifying its clustering separately from the other genotypes.

The predominance of yellow colour in the hierarchical clustering in heatmap format denoted low similarity between the clusters formed (Fig 3). As can also be seen in Fig 3, the uniformity in the yellow coloration for the genetic distances between groups confirms the homogeneity of distances between them.

The variability denoted by the clustering of the accessions corroborates the high estimates of genetic variances and heritabilities displayed by most of the agronomic characteristics; the

total content of fruit pulp carotenoids (TC); and seed characteristics such as mass of seeds per fruit (MSF), ratio of seed to fruit mass (MS/F), and number of seeds per fruit (NSF) (Table 2). This is also analogous to other studies involving the analysis of variability in this crop in Brazil [19, 21].

The greater contribution of the average mass of fruits (MF), diameter of fruit (DF), diameter of internal cavity of fruit (DIC), as well as the mass of seeds per fruit (MSF), and number of seeds per fruit (NSF) for component 1, suggests that there was greater variability for these characteristics, and that they contributed more to genotype discrimination (Fig 5). This result seems to be related to the estimates of genotypic variance, since MSF and NSF also corresponded to characteristics with the greatest genotypic variances (Table 2). The greatest contribution, in component 2, of variables such as the amount of trichomes (AT), leaf recess (LR) and amount of trichomes in the petiole (ATP) shows the importance of multi-categorical characteristics in the discrimination of the studied germplasm.

## Identification of promising groups of genotypes

In *C. moschata*, the identification of promising groups of genotypes can assist in the orientation of crossings targeting hybrid vigour exploitation and the segregation of populations for their characteristics of interest [75, 76].

As shown in Table 4, Group 1 expressed a high genotypic average for total content of fruit pulp carotenoids (TC) and the highest averages for productivity of seeds (PS) and seed oil content (SOC), confirming the high number of promising accessions for these characteristics. The negative correlations between SOC and characteristics related to the quality of fruit pulp in *C. moschata*, such as content of soluble solids (SS) and resistance of fruit pulp to penetration, might hinder simultaneous gains for these characteristics. This can be managed by conducting individualised breeding subprogrammes, aiming in one case to improve seed oil production, and in another, to improve fruit production and quality.

The highest average for total content of fruit pulp carotenoids (TC) occurred in Group 7, formed by the accessions BGH-5455A and BGH-5598A (Table 4). These accessions were also identified as the most promising for TC in the identification *per se*, with new predicted averages greater than 170 μg g$^{-1}$ of fresh pulp mass (Table 5). This result is much higher than those reported in previous studies [4, 37, 77]. Among these, the study involving the characterisation of 55 accessions of *C. moschata*, also maintained by the BGH-UFV, reported a total content of fruit pulp carotenoid averages not greater than 118.70 μg g$^{-1}$ of fresh pulp mass [37]. On the other hand, averages of up to 404.98 μg g$^{-1}$ of fresh pulp mass have been reported [1, 72], when evaluating *C. moschata* germplasm from northeast Brazil. The differences observed for the total content of fruit pulp carotenoids between the present study and previous studies might be mainly associated with the genetic aspects of the germplasm evaluated in each study. According to [72], in northeast Brazil there is a preference for winter squash fruits with more orange pulp, a characteristic associated with higher levels of carotenoids, which corroborates the results obtained for this characteristic in studies involving the evaluation of *C. moschata* germplasm from this region.

Studies with *C. moschata* commonly involve the analysis of fruit pulp carotenoids and generally report high levels of these components [1, 4, 78, 79]. Among these studies, about 19 different carotenoids in the carotenogenic profile of the fruit pulp were identified [1], and *β*- and *α*-carotene constituted the largest proportion of the total carotenoid content in this species. In fact, this vegetable has been considered one of the best sources of carotenoids such as *β*-carotene, with levels above those found in other important carotenogenic vegetables, such as carrots [80].

The main biological functions of components such as $\alpha$- and $\beta$-carotene are their pronounced pro-vitamin A activity [81, 82], and a series of bioactive functions, especially antioxidant activity [83, 84]. Along with its bioactive functions, *C. moschata* brings together fundamental characteristics for biofortification programmes, such as high production potentials and profitability, high efficiency in reducing deficiencies in micronutrients in humans, and good acceptance by producers and consumers in the regions where this crop is grown [8]. *C. moschata* has therefore been strategically used in programmes targeting biofortification in vitamin A precursors, among them the Brazilian Biofortification Programme (BioFORT), led by the Brazilian Agricultural Research Corporation (Embrapa) [9].

The main interest in the assessment of productivity of seeds (PS) and seed oil productivity (SOP) in *C. moschata* corresponds to the high potential for using its seed oil for food purposes. Governments and health experts are interested in encouraging the consumption of unsaturated fatty acids rather than saturated ones, based on the consensus that this reduces the risk of cardiovascular diseases [85, 86, 87], and this vegetable not only has a high oil content, with the lipid fraction of its seeds reaching up to 49% of its composition [88], but the lipid profile of this oil consists of more than 70% unsaturated fatty acids, with a preponderance of fatty acids such as linoleic C18: 2 ($\Delta^{9,12}$) and oleic C18: 1 ($\Delta^{9}$).

*C. moschata* seed oil is also rich in bioactive components such as vitamin E and carotenoids [13], which have important antioxidant activity, in addition to protecting the oil against oxidative processes. Despite this, most of the seeds from the production of *C. moschata* in Brazil are still discarded during consumption. Their use, therefore, represents an alternative way of supplementing diets as well as increasing the income of farmers involved in the production of this vegetable.

Group 16, consisting solely of the control Tetsukabuto, displayed the lowest average for accumulated degree-days for flowering (DDF), indicating that this genotype has the earliest flowering period (Table 4). As can also be seen in the Table 4, most groups had intermediate averages for DDF. Normally, *C. moschata* plants have very long internodes, and this, coupled with the vigorous growth of this species, limits its cultivation, since plants with a greater internode length require much larger areas for cultivation. The interest in assessing precocity in *C. moschata* is based on the possible relationship of this characteristic with aspect such determinate growth habit. According to [89], the *Bu* gene, identified as being responsible for the formation of shorter internodes in pumpkins, is also linked to earlier flowering in this species. In a study evaluating hybrids and segregating winter squash populations for oil production and plant size reduction [50], the cultivars Piramoita and Tronco Verde, which have determinate growth habits, displayed the smallest number of days for female flowering. Greater precocity is an important characteristic for most crops, especially in the cultivation of vegetables, as it optimises the use of cultivation areas, reduces the risks of exposure of the crop to adverse abiotic and biotic factors, and reduces management costs.

In view of the low correlation observed between accumulated degree-days for flowering (DDF) and the other characteristics, it is unlikely that accessions that simultaneously express earlier-flowering and other important characteristics in *C. moschata* will be identified. Therefore, the initial identification of earlier-flowering accessions, followed by incorporation of this trait in germplasm that is promising for other characteristics seems appropriate in *C. moschata* breeding.

Group 4, formed by BGH-1927, BGH-4681A and BGH-5653, had the highest average for productivity of fruits (PF) (Table 4). It also had one of the highest averages for mass of fruits (MF) and an intermediate average for number of fruits per plant (NFP), corroborating the estimates of the correlations between these characteristics and productivity of fruits (Fig 2). The accessions BGH-4681A and BGH-5653 were also identified as the most promising for PF in

the *per se* identification, with averages above 20 t ha$^{-1}$ (Table 5). These averages were much higher than the world average, estimated at 13.4 t ha$^{-1}$ [6].

Although the cultivation of *C. moschata* is primarily intended for fruit production, as already mentioned, the selection of genotypes for greater fruit productivity in this crop must also consider crucial aspects for the acceptability of fruits such as shape and size. In general, winter squash production must currently prioritise the adoption of cultivars with smaller fruits. In addition to obtaining fruits of greater mass, greater productivity in *C. moschata* can also be achieved by obtaining cultivars with higher number of fruits per plant (NFP), based on the estimated correlation observed between productivity of fruits (PF) and NFP (Fig 2).

### *Per se* identification of promising accessions

*Per se* identification of promising accessions can guide selection for a specific trait, allowing the identification of promising accessions for the development of superior inbred lines and/or open-pollinated cultivars. In fact, from a brief survey of the Brazilian National Cultivar Register (RNC), it appears that, of the 182 cultivars of *C. moschata* registered at the moment, most consist of open-pollinated cultivars [74]. This survey also found a considerable number of intra- and interspecific hybrids, confirming the feasibility of applying inbreeding in certain stages of *C. moschata* breeding.

The selected accessions displayed averages for accumulated degree-days for flowering (DDF) much lower than the general averages of the accessions and the controls. Notably, the accessions BGH-6749, BGH-5639, and BGH-219 expressed the lowest new predicted averages for DDF, making them the earliest-flowering accessions (Table 5). Regarding productivity of fruits (PF), the notably more promising accessions were BGH-4453, BGH-5653, BGH-5544A, BGH-4681A, BGH-5224A, and BGH-6587A, which expressed gains above 8 t ha$^{-1}$ and new predicted averages for PF above 20 t ha$^{-1}$ (Table 5). It should be highlighted that the BGH-5544A accession also expressed high averages for productivity of seeds (PS) and seed oil (SOP), corroborating the correlations of these characteristics with productivity of fruits (Fig 2).

The most promising accessions for total content of fruit pulp carotenoids (TC) were BGH-5455A and BGH-5598A (Table 5). These accessions expressed gains and new predicted averages for TC higher than 108.03 and 173.80 µg g$^{-1}$ of fresh pulp mass, respectively, which were much higher than those of the controls. For the characteristics of seed and seed oil, it was found that the accessions BGH-4610A, BGH-5485A, and BGH-6590 were the most promising for productivity of seeds (PS) (Table 6). These accessions expressed gains and new predicted averages for PS of up to 0.31 and 0.58 t ha$^{-1}$, respectively. The most promising accessions for seed oil productivity (SOP) were BGH-5485A, BGH-4610A, and BGH-5472A, which had new predicted averages for SOP of 0.13 t ha$^{-1}$. It is worth highlighting that these accessions corresponded to those with higher PS, corroborating the strong correlation between productivity of seeds and seed oil productivity (Fig 2).

## Conclusions

The accessions of *C. moschata* assessed in this study expressed high genetic variability for agro-morphological characteristics and for agronomic aspects related to the production of seeds such as number and mass of seeds per fruit, for accumulated degree-days for flowering, for total content of fruit pulp carotenoids, and for productivity of fruits, which allowed considerable gains to be obtained from selection for each of these characteristics.

The network of genetic correlations showed that higher fruit productivity in *C. moschata* might be achieved from the selection of aspects considered crucial in the production of this

crop such as higher number of fruits per plant, and height and diameter of fruit. It also showed that greater seed productivity might be achieved with selection for a higher ratio of seed to fruit mass, number and mass of seeds per fruit; this information will assist in selection for higher productivity of fruit, seed and seed oil.

The clustering analysis resulted in 16 groups, with low similarity between the groups, which corroborates the variability of these accessions.

Grouping the averages of the clusters and identification *per se* allowed the most promising groups and accessions to be recognised for each characteristic, an approach that will guide the use of these accessions in breeding programmes.

*Per se* analysis identified the accessions BGH-6749, BGH-5639, and BGH-219 as those with the lowest averages for accumulated degree-days for flowering, highlighting them as the earliest flowering accessions. The most promising accessions for productivity of fruits were BGH-4453, BGH-5653, BGH-5544A, BGH-4681A, BGH-5224A, and BGH-6587A, with new predicted averages greater than 20 t ha$^{-1}$. The accessions with the highest averages for total content of fruit pulp carotenoids were BGH-5455A and BGH-5598A, with averages greater than 170.00 µg g$^{-1}$ of fresh pulp mass. The accessions BGH-5485A, BGH-4610A, and BGH-5472A were the most promising for seed oil productivity, which, in the case of the former two, also corresponded to the highest averages for productivity of seeds. The accessions of *C. moschata* assessed in this study are a promising source for the genetic improvement of characteristics such as early flowering, total content of fruit pulp carotenoids, and productivity of seeds and seed oil.

## Supporting information

**S1 Table. Multi-categorical descriptors used in the assessment of the *C. moschata* germplasm maintained by BGH-UFV.**
(DOCX)

**S2 Table. Result of principal component analysis showing the fifteen principal components and the relative contribution of traits in each component.**
(DOCX)

## Author Contributions

**Conceptualization:** Derly José Henriques da Silva.

**Data curation:** Ronaldo Silva Gomes, Ronaldo Machado Júnior, Cleverson Freitas de Almeida, Rebeca Lourenço de Oliveira, Fabio Teixeira Delazari.

**Investigation:** Ronaldo Silva Gomes, Ronaldo Machado Júnior, Cleverson Freitas de Almeida, Rebeca Lourenço de Oliveira, Fabio Teixeira Delazari.

**Software:** Rafael Ravaneli Chagas.

**Supervision:** Derly José Henriques da Silva.

**Writing – original draft:** Ronaldo Silva Gomes.

**Writing – review & editing:** Ronaldo Silva Gomes, Derly José Henriques da Silva.

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
