## [Decision Letter · Decision Letter 0]

8 Apr 2020

PONE-D-20-05955

Germplasm of Brazilian winter squash (Cucurbita moschata D.) displays vast genetic variability, allowing identification of promising genotypes for agro-morphological traits

PLOS ONE

Dear Mr. Gomes,

Thank you for submitting your manuscript to PLOS ONE. After careful consideration, we feel that it has merit but does not fully meet PLOS ONE’s publication criteria as it currently stands. Therefore, we invite you to submit a revised version of the manuscript that addresses the points raised during the review process.

Address the many points raised by each of the reviewers.  To a large extent these are requests for additional information, clarification, and consistency of terminology and notations.

Some more specific edits to consider -

Add a "Brazil map", information, and pictures described by Reviewer 1. Also, I think this would be helpful in addressing Reviewer 3s' question of the effects of geographic location and environmental conditions.Ensure that the Supplementary file is accessible.Ensure images are sharp and clear (high quality).Add "PCA or PCoA analysis to identify which traits are contributing more information regarding cluster formation."Edit to improve English readability.Do not repeat results in the discussion.

We would appreciate receiving your revised manuscript by May 23 2020 11:59PM. To enhance the reproducibility of your results, we recommend that if applicable you deposit your laboratory protocols in protocols.io, where a protocol can be assigned its own identifier (DOI) such that it can be cited independently in the future. For instructions see: http://journals.plos.org/plosone/s/submission-guidelines#loc-laboratory-protocols

We look forward to receiving your revised manuscript.

Kind regards,

Randall P. Niedz

Academic Editor

PLOS ONE

"This study was financed in part by the Coordenac¸ão de Aperfeic¸oamento de Pessoal deNível Superior – Brasil (CAPES) – Finance Code 001. We also thank the CNPq (National Counsel of Technological and Scientific Development) for the scholarship of the first author."

Reviewers' comments:

Reviewer's Responses to Questions

**Comments to the Author**

1. Is the manuscript technically sound, and do the data support the conclusions?

Reviewer #1: Yes

Reviewer #2: Yes

Reviewer #3: Yes

2. Has the statistical analysis been performed appropriately and rigorously? 

Reviewer #1: Yes

Reviewer #2: Yes

Reviewer #3: Yes

3. Have the authors made all data underlying the findings in their manuscript fully available?

Reviewer #1: Yes

Reviewer #2: Yes

Reviewer #3: Yes

4. Is the manuscript presented in an intelligible fashion and written in standard English?

Reviewer #1: Yes

Reviewer #2: Yes

Reviewer #3: Yes

5. Review Comments to the Author

Reviewer #1: The manuscript by Gomes et al. shows interesting data concerning the agro-morphological evaluation of a wide germplasm collection of Cucurbita moschata. The paper is well conceived, and the experimental procedure is appropriate, nevertheless, I found some issues concerning their manuscript that should be clarified before publication. In addition, I suggest reviewing the organization since many results (data, phrases and reference to tables) are repeated in the discussion section. I recommend a revision of the English form throughout the manuscript.

Regarding Material and methods, I suggest adding a Brazil map with highlighting the different regions from where the accessions derived, or in the Table the authors can add more information as district, province and the coordinates of the area of cultivation just to take note of the geographic distances. It would be nice also to add a figure with pictures of some of the accessions, maybe one from each genetic group or cultivation area, used in the study for showing the most important morphological differences. The authors should provide a justification for the four controls choice. I suppose that the controls are commercial varieties but, the authors should write it and if these varieties are related to some of the accessions used in this study and for what agro-morphological characteristics. Furthermore, the authors should better explain the colorimetric analysis; how many samples used and the meaning of the indexes and formula since are not well explained.

Regarding the Results, the data presented in the supplementary file (that I was not able to download), were not used for the statistical analysis since no explanation of the phytopathogenic resistance of the accessions is reported in the text.

Regarding the discussion, the authors should avoid reporting data, results sentences and tables reference in the discussion. In this way the discussion could be shorter. Furthermore, the powerful of the molecular analysis in identifying duplicates in the germplasm collections and in overcoming the limits of cultivation the Cucurbita moschata accessions (ie. area of cultivation, number of seed), is never mentioned by the authors.

Lane 63: The reference n. 14 does not fit with the sentence

Lane 81: I suggest citing some of the information to which the authors refer to, i.e genetic, agronomic, phenotypic?

Lane 117: of the is repeated twice.

Table 1. The word “Origem” needs to be revised

Lane 195. Authors should change “selection gain” with “gain selection” whose acronym is GS as the authors reported in the following sentence.

Lane 259: The authors are reporting the data of the genotypic variance of all the parameters considered, vegetative, fruits and seeds. Besides to those related to DDF, the not significant data are also those of PS, SOC and SOP. Thus, the sentence should be rewritten. Lane 265. The table 3 caption needs to be simplified

Lane 276 - 279: The sentences are not clear; the data reported in the text are not clearly visible in table 3

Lane 320: The sentence “The red and green lines denote…” in the legend of Figure 1, is in contrast to the sentence in Lane 213 “the dark-green colored lines connect positive-correlated…”, please check.

Lane 341: Why the authors did not used a dendrogram representation of the clustering of the accessions. I think that it could better highlight the accessions similarity among and within the groups.

Lane 349: In the Figure 2 legend, the heatmap coloring should be better explained; yellow color indicates a low similarity, orange color indicates a high similarity

Lane 507. The two controls Jabras and Tetsukabuto that have a similar fruit shape (globular) and fruit weight (2-3 Kg), clustered in two different groups 11 and 16 respectively. Could the authors explain the differences among them and why the Tetsukabuto grouped alone; it could be considered as an outgroup.

Lane 583: The PCOS is never cited in the results

Lane 595: What the authors mean by “genetic makeup”

Lane 750: The title of publication “Priori et al..” is missing

Reviewer #2: Silva Gomes et al. assessed the morphological and genetic diversity of several C. moschata accessions from Brazil using a quantitative genetics approach. They used this information to identify accessions that showed morphological characteristics of agronomic value for promotion of earlier flowering and increase in total carotenoid content and seed oil productivity.

This manuscript is well written and deals with genetic and morphological variation in crops for agronomic improvement. Authors mention that data is available as supplementary information, but I did not have access to supplementary files.

Methods are appropriate, but I suggest adding a PCA or PCoA analysis to identify which traits are contributing more information regarding cluster formation. In addition, the formulas need to be carefully revised for consistency between each term and their intext definition (see specific comments below).

In the results and discussion sections I found it difficult to follow the abbreviation for each trait and I had to go back to table 2 to interpret the results. I recommend using the complete name of each trait with abbreviation between parenthesis the first time they are used in the text.

Line 55. Please change “This has caused the vegetable…” to “This has caused C. moschata…” or “This has caused this vegetable…”

Line 57. Please state the complete name of Embrapa and the abbreviation between parenthesis.

Lines 59-61. Please modify sentence structure so it is clear. Please change to “The seed oil of C. moschata is a good substitute for other lipid sources with higher saturated fatty acid contents, because its seed oil is constituted of about 70% unsaturated fatty acids with a high content of monounsaturated fatty acid [12, 13]”.

Line 68-80. A recent paper by Hernández-Rosales et al. (2020) published in the American Journal of Botany (107(3):510-525) reports high genetic diversity in C. moschata accessions from Mexico and lineage divergence in accordance to altitude. I think you might find it interesting since Mesoamerica has been proposed as one of the possible centers of origin for this species, together with Peru. Also, I recommend the ethnobotanical study by Barrera-Redondo et al. (2020) that examined certain aspects of landrace diversity for C. moschata grown in the central Andes of Peru (Botanical Sciences 98(1):101-116).

Lines 112-116. Please add a figure that shows some of the morphological diversity found in the accessions used in this study.

Line 112. Are the 91 accessions used in the study local landraces?

Line 113. Please estate that control genotypes refer to commercial varieties. Also, explain more thoroughly the experimental design.

Line 132. Please mention the total number of plants and fruits considered in the analysis. Also, the shape of the fruit and peduncle are important traits to characterize the horticultural types of C. moschata. Please mention the diversity of fruit shapes considered in this study.

Line 137. Bioversity International?

Line 153. Please add the abbreviature for total carotenoid (TC) and lutein (L) content in parenthesis.

Line 156. For clarity in the abbreviatures used please differentiate the abbreviation for lutein (L) and luminosity (L).

Lines 195-198. Standardize nomenclature; in example selection gain appears as SG in the text but as GS in the formula. Also, in some formulas you use Pev, while in others you’re using pev. Finally, in formula GS=h2.DS please change the dot by an asterisk to denote multiplication.

Line 200. Check formulas for coefficients of variation because CVg% and CVr% are defined in the same way.

Lines 195-200. In these formulas I see that both genetic and phenotypic variances are incorporated but I do not see how the block effect was incorporated into the analysis.

Line 209. Please check formula, I do not see the term σ2g (y) in it.

Table 3. Please check the range and mean for SOC because the reported mean falls outside the range.

Lines 276-285. Please mention something related to the results for the block effect variance.

Figure 2. Please assign a different colour to each cluster bar. It is very difficult to differentiate between colours.

Table 5. Please add in table caption information regarding the meaning of negative and positive values.

Line 443. Please change “a large areas” for “a large area”.

Lines 485-486. I consider it is important to mention that evethough there is no GWAS for C. moschata, there are genomic analyses for C. pepo. Xanthopoulou et al. (2019; Horticuluture Research 2019(6):94) identified some genes associated to fruit colour and fruit shape in C. pepo; therefore, it is worth mentioning that those genes should also be assessed in C. moschata.

Lines 579-585. I recommend moving this paragraph to results.

Lines 586-599. Regarding the genetic makeup of the germplasm evaluated in this study, how could hybridization (since at least Jabra and Tetsukabuto are hybrids) be influencing the content of carotenoids in the fruit pulp?

Line 696. Please change “the obtainment of” for “obtaining.

Line 703. Please change “The clustering analysis resulted in the formation of 16 groups” for “The clustering analyses resulted in 16 groups”.

Line 705. Please change “the recognition of” for “recognizing”.

Reviewer #3: In this paper, the authors performed an analysis of agro-morphological variation in C. moschata, including relevant characteristics such as earlier-flowering times, carotenoids, seed production, and seed oil productivity. They assessed and compared this variation from BGH-UFV accessions using a thorough experimental design. The results showed correlations and differences among the studied characteristics among accessions, and identified groups of accessions that could help to improve agronomic traits.

I found this paper interesting and properly implemented. The objectives are clear, and the analysis adequate to accomplish them. In general, the manuscript is understandable; however, it needs a style-check to improve readability.

General comments

The figures look very fussy in the pdf; please check the resolution for the final version.

In Methods, the authors should add a justification for all analyses; this will help to understand the analysis rationale to non-specialized readers. For example, what are the purpose of correlation and clustering analyses?

DDF is proposed as a relevant agronomic trait; nevertheless, it did not show a noticeable correlation with another trait (according to Fig. 1). How could this affect the selection of this trait in practice? On the other hand, the seed oil content (SOC) displayed a negative correlation with SS and RP; please include a discussion about the potential trade-offs between traits.

The accessions came from different geographic areas of Brazil. Does this could implicate local adaptation to environmental conditions? How does this potentially influence trait values in practice? Though this is beyond the scope of the paper, the authors should incorporate information from published works about this topic in Discussion.

Specific comments

Lines 156-169. This paragraph is somewhat confusing. “L”, “a” and “b” are defined in line 156, but “L” has a different definition in line 168. Please, clarify.

Line 179 and 226. Please move the reference of from line 226 to 179.

Figure 2. Add a color scale bar for the values, also increase the font of the numbers. As this heatmap represents a square distance matrix, consider removing one of the dendrograms to increase the area of the plot.

Table 7. Change “G” for “g”.

6. PLOS authors have the option to publish the peer review history of their article (what does this mean?). If published, this will include your full peer review and any attached files.

Reviewer #1: Yes: Sara Sestili

Reviewer #2: No

Reviewer #3: No

---

## [Author Response · Author response to Decision Letter 0]

6 May 2020

REBUTTAL LETTER

ANSWERS TO THE POINTS RAISED BY THE ACADEMIC EDITOR

PONE-D-20-05955

Germplasm of Brazilian winter squash (Cucurbita moschata D.) displays vast genetic variability, allowing identification of promising genotypes for agro-morphological traits

PLOS ONE

Dear Mr. Gomes,

Thank you for submitting your manuscript to PLOS ONE. After careful consideration, we feel that it has merit but does not fully meet PLOS ONE’s publication criteria as it currently stands. Therefore, we invite you to submit a revised version of the manuscript that addresses the points raised during the review process.

Address the many points raised by each of the reviewers. To a large extent these are requests for additional information, clarification, and consistency of terminology and notations.

Some more specific edits to consider -

1. Add a "Brazil map", information, and pictures described by Reviewer 1. Also, I think this would be helpful in addressing Reviewer 3s' question of the effects of geographic location and environmental conditions.

Answer: We find the suggestion quite pertinent. We found more appropriated adding a Brazilian map highlighting the different states from where the accessions derived. In the reviewed version, the map corresponds to the lines 116-117.

2. Ensure that the Supplementary file is accessible

Answer: all the supplementary files were attached this time.

3. Ensure images are sharp and clear (high quality).

Answer: They have been improved and are much sharper and clearer (higher quality/definition). Figure 1 (lines 116-117), Figure 2 (lines 323-324), Figure 3 (lines 371-372), Figure 4 (lines 383-384), and Figure 5 (lines 396-397). 

4. Add "PCA or PCoA analysis to identify which traits are contributing more information regarding cluster formation."

Answer: we found quite pertinent adding this analysis and we opted for a PCA analysis. The results of this analysis were added in the lines (389-396), and the discussion in the lines (636-644).

5. Edit to improve English readability.

Answer: Concerning the English revision , we would like to mention that the manuscript was carefully revised by a professional service, the English language editing services for Academic, Scientific Manuscripts, Articles and Papers (Editage- https://www.editage.com/). We took a close look in the considerations raised in the English review before sending the manuscript to Plos One. After making all the arrangements suggested by the editor and reviewers, we sent the manuscript for a second English review. We are sending the reviewing certificates attesting both reviews. We hope having fulfilled the requirements in terms of the English writing and style.

6. Do not repeat results in the discussion.

Answer: I recognized the need for reviewing/improving the organization of the manuscript version submitted to Plos One. It was true that much of the results were repeated in the discussion. We removed the results from the discussion and made considerable rearrangement/improvement concerning this.

We would appreciate receiving your revised manuscript by May 23 2020 11:59PM. To enhance the reproducibility of your results, we recommend that if applicable you deposit your laboratory protocols in protocols.io, where a protocol can be assigned its own identifier (DOI) such that it can be cited independently in the future. For instructions see: http://journals.plos.org/plosone/s/submission-guidelines#loc-laboratory-protocols

• A rebuttal letter that responds to each point raised by the academic editor and reviewer(s). This letter should be uploaded as separate file and labeled 'Response to Reviewers'.

• A marked-up copy of your manuscript that highlights changes made to the original version. This file should be uploaded as separate file and labeled 'Revised Manuscript with Track Changes'.

• An unmarked version of your revised paper without tracked changes. This file should be uploaded as separate file and labeled 'Manuscript'.

We look forward to receiving your revised manuscript.

Kind regards,

Randall P. Niedz

Academic Editor

PLOS ONE

"This study was financed in part by the Coordenac¸ão de Aperfeic¸oamento de Pessoal deNível Superior – Brasil (CAPES) – Finance Code 001. We also thank the CNPq (National Counsel of Technological and Scientific Development) for the scholarship of the first author."

Please provide an amended statement that declares *all* the funding or sources of support (whether external or internal to your organization) received during this study, as detailed online in our guide for authors at http://journals.plos.org/plosone/s/submit-now. Please also include the statement “There was no additional external funding received for this study.” in your updated Funding Statement. Please include your amended Funding Statement within your cover letter. We will change the online submission form on your behalf.

Answer: we added the following text: All the funding sources of support to this study corresponded to study scholarships received by the first author (Ronaldo Silva Gomes). The first scholarship corresponded to a master's scholarship (grant number 001), funded by the Coordination for the Improvement of Higher Education Personnel (CAPES). The second scholarship corresponded to a doctorate scholarship (doctorate-GD grant), funded by the National Council of Technological and Scientific Development (CNPq). There was no additional external funding received for this study.” in your updated Funding Statement. 

Reviewers' comments:

Reviewer's Responses to Questions

Comments to the Author

1. Is the manuscript technically sound, and do the data support the conclusions?

Reviewer #1: Yes

Reviewer #2: Yes

Reviewer #3: Yes

 2. Has the statistical analysis been performed appropriately and rigorously?

Reviewer #1: Yes

Reviewer #2: Yes

Reviewer #3: Yes

3. Have the authors made all data underlying the findings in their manuscript fully available?

Reviewer #1: Yes

Reviewer #2: Yes

Reviewer #3: Yes

4. Is the manuscript presented in an intelligible fashion and written in standard English?

Reviewer #1: Yes

Reviewer #2: Yes

Reviewer #3: Yes

5. Review Comments to the Author

Please use the space provided to explain your answers to the questions above. You may also include additional comments for the author, including concerns about dual publication, research ethics, or publication ethics. (Please upload your review as anattachment if it exceeds 20,000 characters)

REVIEWER #1

Reviewer #1: The manuscript by Gomes et al. shows interesting data concerning the agro-morphological evaluation of a wide germplasm collection of Cucurbita moschata. The paper is well conceived, and the experimental procedure is appropriate, nevertheless, I found some issues concerning their manuscript that should be clarified before publication. In addition, I suggest reviewing the organization since many results (data, phrases and reference to tables) are repeated in the discussion section. I recommend a revision of the English form throughout the manuscript.

Answer: I recognized the need for reviewing/improving the organization of the manuscript version submitted to Plos One. It was true that much of the results were repeated in the discussion. We removed the results from the discussion and made considerable rearrangement/improvement concerning this.

Concerning the English revision recommended by the reviewer #1, we would like to mention that the manuscript was carefully revised by a professional service, the English language editing services for Academic, Scientific Manuscripts, Articles and Papers (Editage- https://www.editage.com/). We took a close look in the considerations raised in the English review before sending the manuscript to Plos One. After making all the arrangements suggested by the editor and reviewers, we sent the manuscript for a second English review. We are sending the reviewing certificates attesting both reviews. We hope having fulfilled the requirements in terms of the English writing and style. 

Reviewer #1: Regarding Material and methods, I suggest adding a Brazil map with highlighting the different regions from where the accessions derived, or in the Table the authors can add more information as district, province and the coordinates of the area of cultivation just to take note of the geographic distances. 

Answer: We find the suggestion quite pertinent. We found more appropriated adding a Brazilian map highlighting the different states from where the accessions derived. In the reviewed version, the map corresponds to the lines 116-117.

Reviewer #1: It would be nice also to add a figure with pictures of some of the accessions, maybe one from each genetic group or cultivation area, used in the study for showing the most important morphological differences. 

Answer: Reviewer #2 also raised this point. We find the suggestion quite pertinent. We added a figure with representative fruits of some of the largest groups formed in the clustering analysis. The figure also has representative fruits of some small groups formed in the clustering analysis and show important morphological differences between the fruits of different groups. In the reviewed version, the figure corresponds to the lines 383-384. 

Reviewer #1: The authors should provide a justification for the four controls choice. I suppose that the controls are commercial varieties but, the authors should write it and if these varieties are related to some of the accessions used in this study and for what agro-morphological characteristics. 

Answer: Reviewer#2 also raised this point. The controls were chosen because they consist of cultivars widely cultivated and commercialized in Brazil. Thus we believed that they would fit as good standards for comparing the accessions. In the reviewed version, the clarification regarding this point corresponds to the lines 112-114. 

Reviewer #1: Furthermore, the authors should better explain the colorimetric analysis; how many samples used and the meaning of the indexes and formula since are not well explained.

Answer: as we mention in the manuscript, the estimates of the total content of fruit pulp carotenoids (TC) and lutein (L) were obtained based on colorimetric parameters. For this, the fruit pulp colour was characterised with the aid of a manual tri-stimulus colorimeter, Colour Reader CR-10 Konica Minolta, based on reading of the parameters related to luminosity, and contribution of red (a) and yellow (b). The characterization of fruit pulp was made from a fruit from each of the three central plants of the plot. This was carried out from four different regions of the fruit pulp (region facing the sun, region facing the soil, region of the peduncle and floral insertion). Thus, the values of each parameter consisted of averages obtained from readings of the pulp of fruits harvested from each of the plot central plants. In the reviewed version, the clarification regarding this point corresponds to the lines 151-159. 

Reviewer #1: Regarding the Results, the data presented in the supplementary file (that I was not able to download), were not used for the statistical analysis since no explanation of the phytopathogenic resistance of the accessions is reported in the text.

Answer: all the supplementary files were attached this time. Yes, there was a mistake because we mentioned the trait “phytopathogenic resistance”, which we did not evaluated. We made the correction and the term is no longer mentioned in the text. We must clarify that supplementary file is a table and contains the description of the qualitative descriptors also used in the germplasm assessment. The data described in supplementary file was used in the statistical analysis. As we explain in the lines (217-249), this data set was used in the analysis of variability. 

Reviewer #1: regarding the discussion, the authors should avoid reporting data, results sentences and tables reference in the discussion. In this way the discussion could be shorter. Furthermore, the powerful of the molecular analysis in identifying duplicates in the germplasm collections and in overcoming the limits of cultivation the Cucurbita moschata accessions (ie. area of cultivation, number of seed), is never mentioned by the authors. (phenotyping in the field…)

Answer: I recognized the need for reviewing/improving the organization of the manuscript. We removed the results from the discussion and made considerable rearrangement/improvement concerning this. We believe that the discussion much concise after reviewing/improving this point. 

We recognize the great usefulness of molecular analysis in the management of plant genetic resources. At the same, the field-phenotyping of C. moschata is crucial in the assessment of this crop, which lead us focusing in the assessment of the germplasm in the field. 

Reviewer #1: Lane 63: The reference n. 14 does not fit with the sentence.

Answer: We recognized the mistake and made the correction. Now the association between this citation and its reference is correct. We also took a double check in all the citations and their respective references and made sure that all of them were correct. 

Reviewer #1: Lane 81: I suggest citing some of the information to which the authors refer to, i.e genetic, agronomic, phenotypic?

Answer: we are referring to genetic and phenotypic data. We have made the clarification in the lines 81-83. 

Reviewer #1: Lane 117: of the is repeated twice.

Answer: the correction has been made. 

Reviewer #1: Table 1. The word “Origem” needs to be revised

Answer: the table was replaced by the figure 1 and the word origem was revised. 

Reviewer #1: Lane 195. Authors should change “selection gain” with “gain selection” whose acronym is GS as the authors reported in the following sentence.

Answer: Reviewer #2 also suggested us standardizing the nomenclature through the text. We made a careful revision and correction regarding this point. 

Reviewer #1: Lane 259: The authors are reporting the data of the genotypic variance of all the parameters considered, vegetative, fruits and seeds. Besides to those related to DDF, the not significant data are also those of PS, SOC and SOP. Thus, the sentence should be rewritten. 

Answer: The subtitle was improved and changed to” Variance components and genetic-statistical parameters of the agronomic aspects, total content of fruit pulp carotenoids, and the characteristics of seeds and seed oil”. We believe that the subtitle is more appropriate and informative now.

When we mention, “Among these variance estimates, only the genotypic variance of DDF was not significant”, we are referring strictly to the traits mentioned in the last paragraph - the number of seeds (NSF), mass of seeds per fruit (MSF), to the degree-days accumulated for flowering (DDF), and to the total content of fruit pulp carotenoids (TC). In the reviewed version, this corresponds to the lines 261-266. 

Reviewer #1: Lane 265. The table 3 caption needs to be simplified.

Answer: The correction has been made and the caption is more concise now. The reviewed version corresponds to the line 267.

Reviewer #1: Lane 276 - 279: The sentences are not clear; the data reported in the text are not clearly visible in table 3.

Answer: we realized that the sentences were not clear. We simplified the information which is much clearer now. In the reviewed version this corresponds to the lines 277-280. 

Reviewer #1: Lane 320: The sentence “The red and green lines denote…” in the legend of Figure 1, is in contrast to the sentence in Lane 213 “the dark-green colored lines connect positive-correlated…”, please check.

Answer: we realized the mistake and corrected it. The reviewed version corresponds to the lines 210-213 and 324-335.

Reviewer #1: Lane 341: Why the authors did not used a dendogram representation of the clustering of the accessions. I think that it could better highlight the accessions similarity among and within the groups. 

Answer: as we describe in the section material and methods, we implemented the clustering using the method called Affinity, proposed by Frey BJ, Dueck D. Clustering by passing messages between data points. Science. 2007; doi: 10.1126/science.1136800. 

Affinity corresponds to a method widely used in the analysis of clustering, including the clustering of plant germplasm. The clustering output from Affinity consists of two informations: a) the discriminations of the groups formed and of the individuals allocated to each group, and b) a heatmap and a hierarchical clustering showing the distances between each pair of genotype. As can be visualized in the figure 3, the heat map also consist of a dendogram. The left axis shows, in a dendogram format, the 16 groups formed in the clustering analysis.

We wanted to show the heatmap in order display the distances between the groups. If we had opted to show only the heatmap, it would be difficult discriminating each accession, thus we found more appropriated adding a table (Table 3), discriminating the groups and their respective accessions resulting from the clustering. 

Reviewer #1: Lane 349: In the Figure 2 legend, the heatmap coloring should be better explained; yellow color indicates a low similarity, orange color indicates a high similarity. 

Answer: we identified the need of clarifying the figure legend. In the reviewed version, this corresponds to the lines 372-378.

Reviewer #1: Lane 507. The two controls Jabras and Tetsukabuto that have a similar fruit shape (globular) and fruit weight (2-3 Kg), clustered in two different groups 11 and 16 respectively. Could the authors explain the differences among them and why the Tetsukabuto grouped alone; it could be considered as an outgroup.

Answer: we found pertinent adding a brief discussion regarding the clustering of Jabras and Tetsukabuto. We added the following discussion: It is notable that the two hybrids Jabras and Tetsukabuto used as controls clustered in different groups. Although they have similar fruit shape and size, the groups to which they were allocated differ for most of the characteristics (Table 4). In addition, these cultivars expressed different genotypic values for most of the characteristics (Tables 5 and 6), justifying their clustering in different groups. Tetsukabuto is an interspecific hybrid between C. moschata and C. maxima [73]; corresponded to the group with lowest genotypic average for degree-days accumulated for flowering (DDF), in addition to expressing genotypic averages quite different from the other groups in relation to the characteristics of seeds and seed oil (Table 4), justifying its clustering apart from the other genotypes. The reviewed version corresponds to the lines 618-626.

Reviewer #1: Lane 583: The PCOS is never cited in the results.

Answer: we realized the mistake and corrected it. Actually the term referred to seed oil productivity (SOP).

Reviewer #1: Lane 595: What the authors mean by “genetic makeup”.

Answer: by mentioning “genetic makeup” we meant genetic aspects. The reviewed version corresponds to the line 667.

Reviewer #1: Lane 750: The title of publication “Priori et al..” is missing.

Answer: we realized the mistake and added the title. We also took a double check in all the citations and their respective references and made sure that all of them were correct. 

REVIEWER #2:

Reviewer #2: Silva Gomes et al. assessed the morphological and genetic diversity of several C. moschata accessions from Brazil using a quantitative genetics approach. They used this information to identify accessions that showed morphological characteristics of agronomic value for promotion of earlier flowering and increase in total carotenoid content and seed oil productivity.

Reviewer #2: This manuscript is well written and deals with genetic and morphological variation in crops for agronomic improvement. Authors mention that data is available as supplementary information, but I did not have access to supplementary files.

Answer: all the supplementary files were added this time.

Reviewer #2: Methods are appropriate, but I suggest adding a PCA or PCoA analysis to identify which traits are contributing more information regarding cluster formation. In addition, the formulas need to be carefully revised for consistency between each term and their intext definition (see specific comments below).

Answer: we found quite pertinent adding this analysis and we opted for a PCA analysis. The results of this analysis were added in the lines (389-396), and the discussion in the lines (636-644)

Reviewer #2: In the results and discussion sections I found it difficult to follow the abbreviation for each trait and I had to go back to table 2 to interpret the results. I recommend using the complete name of each trait with abbreviation between parenthesis the first time they are used in the text.

Answer: we recognize the difficult of following the abbreviations through the text. We found more appropriate wrintiying the full term followed by its abbreviation if was the first time the term was mentioned in a paragraph. If the term has been mentioned in a paragraph once, in the second time we refer to it in the same paragraph by mentioning its abbreviation. We believe that the text is clearer this way and is also concise. We kept this pattern through the whole text. 

Reviewer #2: Line 55. Please change “This has caused the vegetable…” to “This has caused C. moschata…” or “This has caused this vegetable…”

Answer: the change has been made. The reviewed version corresponds to the line 55.

Reviewer #2: Line 57. Please state the complete name of Embrapa and the abbreviation between parenthesis.

Answer: the change has been made. The reviewed version corresponds to the lines 57.

Reviewer #2: Lines 59-61. Please modify sentence structure so it is clear. Please change to “The seed oil of C. moschata is a good substitute for other lipid sources with higher saturated fatty acid contents, because its seed oil is constituted of about 70% unsaturated fatty acids with a high content of monounsaturated fatty acid .

Answer: the change has been made. The reviewed version corresponds to the lines 59-67.

Reviewer #2: Line 68-80. A recent paper by Hernández-Rosales et al. (2020) published in the American Journal of Botany (107(3):510-525) reports high genetic diversity in C. moschata accessions from Mexico and lineage divergence in accordance to altitude. I think you might find it interesting since Mesoamerica has been proposed as one of the possible centers of origin for this species, together with Peru. Also, I recommend the ethnobotanical study by Barrera-Redondo et al. (2020) that examined certain aspects of landrace diversity for C. moschata grown in the central Andes of Peru (Botanical Sciences 98(1):101-116).

Answer: I read both papers and realized that the first study, of Hernández-Rosales et al. 2020, brings important information regarding the eco-geographical distribution of C. moschata germplasm in Mexico. The study of Hernández-Rosales et al. 2020 corroborates some of our results, especially the variability in C. moschata germplasm and has been added in the citations. 

Reviewer #2: Lines 112-116. Please add a figure that shows some of the morphological diversity found in the accessions used in this study.

Answer: We find the suggestion quite pertinent. We added a figure with representative fruits of some of the largest groups formed in the clustering analysis. The figure also has representative fruits of some small groups formed in the clustering analysis and show important morphological differences between the fruits of different groups (Figure 4).

Reviewer #2: Line 112. Are the 91 accessions used in the study local landraces?

Answer: yes, they are local landraces. We clarified this and the reviewed version corresponds to the lines 114-116.

Reviewer #2: Line 113. Please estate that control genotypes refer to commercial varieties. Also, explain more thoroughly the experimental design.

Answer: The controls were chosen because they consist of cultivars widely cultivated and commercialized in Brazil. Thus we believed that they would fit as good standards for comparing the accessions. The reviewed version corresponds to the lines 112-114.

Regarding the experimental design, the details are the section “Experiment location and experimental design” which corresponds to the lines 123-132.

Reviewer #2: Line 132. Please mention the total number of plants and fruits considered in the analysis. Also, the shape of the fruit and peduncle are important traits to characterize the horticultural types of C. moschata. Please mention the diversity of fruit shapes considered in this study. 

Answer: the information concerning the total number of plants and fruits considered in the analysis was clarified. The reviewed version concerning this point corresponds to the lines 130-132.

Reviewer #2: Also, the shape of the fruit and peduncle are important traits to characterize the horticultural types of C. moschata. Please mention the diversity of fruit shapes considered in this study.

Answer: the traits shape of the fruit and peduncle were considered in the assessment of the germoplasm and are described in the supplementary file (Supplementary Table 1). The diversity of fruit shapes is discussed in the lines 558-565.

Reviewer #2: Line 137. Bioversity International?

Answer: yes, we meant Bioversity International and the correction has been made. In the reviewed version the correct form corresponds to the lines 136.

Reviewer #2: Line 153. Please add the abbreviation for total carotenoid (TC) and lutein (L) content in parenthesis.

Answer: We found more appropriate wrintiying the full term followed by its abbreviation if was the first time the term was mentioned in a paragraph. If the term has been mentioned in a paragraph once, in the second time we refer to it in the same paragraph by mentioning its abbreviation. We believe that the text is clearer this way and is also concise. We kept this pattern through the whole text. 

Reviewer #2: Reviewer#3 also raised this point. Line 156. For clarity in the abbreviations used please differentiate the abbreviation for lutein (L) and luminosity (L).

Answer: we realized that the luminosity is not used in the equations for the estimation of the total content of fruit pulp carotenoids, so the is no need of abbreviating the term luminosity. In the reviewed version the correct form corresponds to the lines 151-154.

Reviewer #2: Lines 195-198. Standardize nomenclature; in example selection gain appears as SG in the text but as GS in the formula. Also, in some formulas you use Pev, while in others you’re using pev. Finally, in formula GS=h2.DS please change the dot by an asterisk to denote multiplication.

Answer: We realized the mistakes. The corrections has been made. In the reviewed version the correct form corresponds to the lines 191-199.

Reviewer #2: Line 200. Check formulas for coefficients of variation because CVg% and CVr% are defined in the same way.

Answer: We realized the mistake. The corrections has been made. In the reviewed version the correct form corresponds to the lines 199.

Reviewer #2: Lines 195-200. In these formulas I see that both genetic and phenotypic variances are incorporated but I do not see how the block effect was incorporated into the analysis.

Answer: from the genetic-statistical parameters estimates obtained in this study, the block effect is usually incorporated only in the estimate of heritability. In this study we found more appropriate estimating heritability based on the prediction error variance (Pev), an approach proposed by (Cullis et al., 2006 - Cullis BR, Smith AB, Coombes NE. On the design of early generation variety trials with correlated data. J Agric Biol Envir S. 2006; doi: 10.1198/108571106X154443).

Reviewer #2: Line 209. Please check formula, I do not see the term σ2g (y) in it.

Answer: The corrections has been made. In the reviewed version the correct form corresponds to the lines 203.

Reviewer #2: Table 3. Please check the range and mean for SOC because the reported mean falls outside the range.

Answer: The corrections has been made. In the reviewed version the correct form corresponds to the lines 301-304.

Reviewer #2: Lines 276-285. Please mention something related to the results for the block effect variance.

Answer: the result regarding the block effect variance was mentioned. In the reviewed version the correct form corresponds to the lines 265-266.

Reviewer #2: Figure 2. Please assign a different colour to each cluster bar. It is very difficult to differentiate between colours. (I believe it has been improved since the resolution figure resolution was improved). 

Answer: All the figures have been improved and have a higher definition now. Thus we believe it will be easier differentiating between the colours of the cluster bars now. 

Reviewer #2: Table 5. Please add in table caption information regarding the meaning of negative and positive values.

Answer: the information has been added. In the reviewed version the correct form corresponds to the lines 437-438.

Reviewer #2: Line 443. Please change “a large areas” for “a large area”.

Answer: the change has been made. In the reviewed version the correct form corresponds to the lines 510.

Reviewer #2: Lines 485-486. I consider it is important to mention that evethough there is no GWAS for C. moschata, there are genomic analyses for C. pepo.; Horticuluture Research 2019(6):94) identified some genes associated to fruit colour and fruit shape in C. pepo; therefore, it is worth mentioning that those genes should also be assessed in C. moschata.

Answer: I read the study of Xanthopoulou et al. (2019) and I could not see a straight relationship with our study. 

Reviewer #2: Lines 579-585. I recommend moving this paragraph to results.

Answer: the change has been made. 

Reviewer #2: Lines 586-599. Regarding the genetic makeup of the germplasm evaluated in this study, how could hybridization (since at least Jabra and Tetsukabuto are hybrids) be influencing the content of carotenoids in the fruit pulp? 

Answer: As far as we know, there is no information about the influence of hybridization on the carotenoid content in winter squash fruits. We also intend to study, the genetic control of the carotenoid content of fruit pulp and the content of carotenoids such as β and α-carotene in C. moschata.

Reviewer #2: Line 696. Please change “the obtainment of” for “obtaining.

Answer: the change has been made. In the reviewed version the correct form corresponds to the lines 768.

Reviewer #2: Line 703. Please change “The clustering analysis resulted in the formation of 16 groups” for “The clustering analyses resulted in 16 groups”.

Answer: the change has been made. In the reviewed version the correct form corresponds to the lines 775.

Reviewer #2: Line 705. Please change “the recognition of” for “recognizing”.

Answer: the change has been made. In the reviewed version the correct form corresponds to the lines 778.

REVIEWER #3:

Reviewer #3: In this paper, the authors performed an analysis of agro-morphological variation in C. moschata, including relevant characteristics such as earlier-flowering times, carotenoids, seed production, and seed oil productivity. They assessed and compared this variation from BGH-UFV accessions using a thorough experimental design. The results showed correlations and differences among the studied characteristics among accessions, and identified groups of accessions that could help to improve agronomic traits.

Reviewer #3: I found this paper interesting and properly implemented. The objectives are clear, and the analysis adequate to accomplish them. In general, the manuscript is understandable; however, it needs a style-check to improve readability.

Answer: Concerning the English revision recommended by the reviewer #1, we would like to mention that the manuscript was carefully revised by a professional service, the English language editing services for Academic, Scientific Manuscripts, Articles and Papers (Editage- https://www.editage.com/). We took a close look in the considerations raised in the English review before sending the manuscript to Plos One. After making all the arrangements suggested by the editor and reviewers, we sent the manuscript for a second English review. We are sending the reviewing certificates attesting both reviews. We hope having fulfilled the requirements in terms of the English writing and style. 

Reviewer #3: General comments

Reviewer #3: The figures look very fussy in the pdf; please check the resolution for the final version.

Answer: All the figures have been improved and have a higher definition now. Thus we believe it will be easier visualizing the figures informations. 

Reviewer #3: In Methods, the authors should add a justification for all analyses; this will help to understand the analysis rationale to non-specialized readers. For example, what are the purpose of correlation and clustering analyses?

Answer: all the analysis are justified in the beginning of their respective sections in the discussion. 

Reviewer #3: DDF is proposed as a relevant agronomic trait; nevertheless, it did not show a noticeable correlation with another trait (according to Fig. 1). How could this affect the selection of this trait in practice? On the other hand, the seed oil content (SOC) displayed a negative correlation with SS and RP; please include a discussion about the potential trade-offs between traits.

Answer: we found appropriate adding a brief discussion regarding these results. “In view of the low correlation observed between degree-days accumulated for flowering (DDF) and the other characteristics, it is unlikely to identify accessions that simultaneously express earlier-flowering and other important characteristics in C. moschata. With this, the initial identification of earlier-flowering accessions, followed by the incorporation of these trait in promising germplasm for other characteristics, seems appropriate in C. moschata breeding”. In the reviewed version the correct form corresponds to the lines 716-720.

“The negative correlations between SOC and characteristics related to the quality of fruit pulp in C. moschata such as content of soluble solids (SS) and resistance of fruit pulp to penetration observed in this study might hinder simultaneous gains for these characteristics. This can be managed by conducting individualized subprograms breeding, aiming in one case to improve seed oil production, in another, to improve fruit production and quality”. In the reviewed version the correct form corresponds to the lines 651-656.

Reviewer #3: The accessions came from different geographic areas of Brazil. Does this could implicate local adaptation to environmental conditions? How does this potentially influence trait values in practice? Though this is beyond the scope of the paper, the authors should incorporate information from published works about this topic in Discussion.

Answer: we found pertinent adding a brief discussion regarding this point. “The clustering did not reflect a smaller genetic distance between those accessions from the same state or geographic region of Brazil. Group 11, for example, grouped accessions from different states and regions; and the preponderance of accessions from Minas Gerais (MG) and São Paulo (SP) in this group was probable only result of the greater number of accessions from these states. This trend was repeated for other groups with higher numbers of accessions such as 1, 5 and 14. In a study involving the assessment of C. moschata accessions from different regions of Brazil and maintained at BGH-UFV, [73] also did not report smaller genetic distance between the accessions from the same state or region. In the reviewed version the correct form corresponds to the lines 610-617.

Reviewer #3: Specific comments

Reviewer #3: Lines 156-169. This paragraph is somewhat confusing. “L”, “a” and “b” are defined in line 156, but “L” has a different definition in line 168. Please, clarify.

Answer: we realized that the luminosity is not used in the equations for the estimation of the total content of fruit pulp carotenoids, so the is no need of abbreviating the term luminosity. In the reviewed version the correct form corresponds to the lines 152-154.

Reviewer #3: Line 179 and 226. Please move the reference of from line 226 to 179.

Answer: the citation in the line 179 is correct. We corrected the citation style in the line 226. In the reviewed version the correct form corresponds to the line 223.

Reviewer #3: Figure 2. Add a color scale bar for the values, also increase the font of the numbers. As this heatmap represents a square distance matrix, consider removing one of the dendrograms to increase the area of the plot.

Answer: the bar has been added. All the figures have been improved and have a higher definition now. Thus we believe it will be easier visualizing the figures informations.

Reviewer #3: Table 7. Change “G” for “g”.

Answer: the change has been made (lines 487-488). 

6. PLOS authors have the option to publish the peer review history of their article (what does this mean?). If published, this will include your full peer review and any attached files.

Do you want your identity to be public for this peer review? For information about this choice, including consent withdrawal, please see our Privacy Policy.

Reviewer #1: Yes: Sara Sestili

Reviewer #2: No

Reviewer #3: No

---

## [Editor Report · Decision Letter 1]

15 May 2020

Brazilian germplasm of winter squash (Cucurbita moschata D.) displays vast genetic variability, allowing identification of promising genotypes for agro-morphological traits

PONE-D-20-05955R1

Dear Dr. Gomes,

Good job on the revision.

We are pleased to inform you that your manuscript has been judged scientifically suitable for publication and will be formally accepted for publication once it complies with all outstanding technical requirements.

With kind regards,

Randall P. Niedz

Academic Editor

PLOS ONE
---

## [Editor Report · Acceptance letter]

21 May 2020

PONE-D-20-05955R1 

Brazilian germplasm of winter squash (*Cucurbita moschata* D.) displays vast genetic variability, allowing identification of promising genotypes for agro-morphological traits     

Dear Dr. Gomes:

I am pleased to inform you that your manuscript has been deemed suitable for publication in PLOS ONE. Congratulations! Your manuscript is now with our production department. 

With kind regards,

on behalf of

Dr. Randall P. Niedz 

Academic Editor

PLOS ONE